# Disruption of the SAGA CORE triggers collateral degradation of KAT2A

Paul Batty [1,2], Hannah Beneder[1,2], Caroline Schätz [2], Gabriel Onea[2,3], Maciej Zaczek [1,2], Ana P. Kutschat [1,2], Miriam Abele[2], Sophie Müller [1,2], Giulio Superti-Furga [2,4], Georg E. Winter [2,5] & Davide Seruggia [1,2] ✉

The Spt-Ada-Gcn5 acetyltransferase (SAGA) complex regulates gene expression through histone acetylation at promoters, mediated by its histone acetyl transferase (HAT), KAT2A. While SAGA structure and function are well characterised, mechanisms controlling the stability of individual subunits, including KAT2A, remain unclear. Here, using a fluorescence-based KAT2A stability reporter, we systematically dissect the molecular dependencies controlling KAT2A protein abundance, and identify the non-enzymatic SAGA CORE module subunits—TADA1, TAF5L, and TAF6L— as necessary for KAT2A stability. Loss of these subunits disrupts SAGA complex integrity, leading to non-chromatin-bound KAT2A that is degraded by the proteasome and consequent reduced H3K9 acetylation. Proteomic profiling reveals progressive loss of components from the CORE and HAT modules upon acute SAGA CORE disruption, indicating that an intact CORE is required for the stability of numerous SAGA components. Finally, a focused CRISPR screen of ubiquitin-proteasome system genes identifies the E3 ligase UBR5, a known regulator of orphan protein degradation, and the deubiquitinase OTUD5, as regulators of KAT2A degradation when the SAGA CORE is perturbed. Together, these findings reveal a dependency of KAT2A protein stability on SAGA CORE integrity and define an orphan quality control mechanism targeting unassembled KAT2A, revealing a potential vulnerability in SAGA-driven malignancies.

Gene expression is controlled by large multi-protein complexes that interact with chromatin and transcription factors to facilitate or repress transcription[1]. The biogenesis and maintenance of such multi-protein complexes is a highly regulated process[2,3], with ribonucleoprotein immunoprecipitation assays demonstrating that the assembly of large protein complexes is often co-translational[3,4]. For instance, components of COMPASS[5], TFIID[6] and other co-activator complexes, including ATAC and SAGA[7], display paired interactions in the cytosol already at the stage of nascent protein, suggesting that components of these complexes are assembled co-translationally. This mechanism functions to extend the half-life of intrinsically unstable components, with the partner protein acting as a chaperone for aggregation-prone subunits.

Numerous quality control mechanisms exist to maintain homeostasis in protein complex assembly. These mechanisms are crucial to ensure that proteins fold correctly and are expressed in the right amounts, thereby preventing the formation of mislocalised, misfolded, or aggregated proteins, which would have deleterious consequences for the cell[8,9]. However, as protein complex assembly is not a stoichiometric process and complex partners have different abundancies in the cell, some subunits remain uncomplexed as there are insufficient binding partners to successfully form the full complex.

[1]St. Anna Children's Cancer Research Institute (CCRI), Vienna, Austria. [2]CeMM Research Center for Molecular Medicine of the Austrian Academy of Sciences, Vienna, Austria. [3]Department of Pediatrics and Adolescent Medicine, Medical University of Vienna, Vienna, Austria. [4]Center for Physiology and Pharmacology, Medical University of Vienna, Vienna, Austria. [5]AITHYRA Institute for Biomedical Artificial Intelligence, Vienna, Austria. ✉e-mail: davide.seruggia@ccri.at

These orphan proteins may become bound by chaperones until such a time that a partner protein becomes available, but in the absence of their normal partners can also misfold. Misfolding can, in turn, expose hidden degrons or hydrophobic patches that ordinarily are inaccessible in the assembled complex[10]. These de-novo degrons then mediate protein degradation via specialised machinery of the orphan quality control system, thereby maintaining protein homeostasis[11–13].

Interestingly, inherent features of these two mechanisms, i.e. dependency on co-translational assembly and dedicated orphan quality control, can be exploited using chemical or genetic approaches. For example, chemical or genetic[14–16] perturbation of the PRC2 component EED leads to loss of its complex partners, EZH2 and SUZ12, in a phenomenon termed collateral degradation[17]. Similarly, PROTACs targeting HDAC1/2 result in the degradation of other proteins in the same complex[18], such as LSD1 of CoREST, or components of the SIN3 complex[19], while genetic depletion of HDAC1/2 in neuroblastoma cells leads to destabilisation of NuRD[17]. Such collateral degradation of partner proteins upon loss of a single complex component has emerged as a new therapeutic modality when targeting co-activator or co-repressor complexes that are deregulated in cancer[20], as specific targeting of one subunit has the potential to result in collateral loss of multiple subunits and consequent loss of complex activity.

The SAGA (Spt-Ada-Gcn5 acetyltransferase) is a large co-activator complex composed of 20 proteins organised into five distinct modules[21]. Two of these five modules, the DUB (Deubiquitinase) and HAT (Histone Acetyltransferase), have enzymatic activity, conferred by the histone deubiquitinase USP22, which deubiquitinates Lysine 120 of Histone H2B, and the acetyltransferase KAT2A (also known as GCN5)[22], which predominantly acetylates Histone H3K9 at promoters to facilitate transcriptional activation. KAT2A has a paralogue, KAT2B (also known as PCAF), which shares >70% sequence homology with KAT2A, with the two acetyltransferases mutually exclusively incorporated into the SAGA HAT module[23,24]. The other three modules of SAGA lack enzymatic activity and have various functions, from mediating interactions with transcription factors (TRRAP), associating with splicing and DNA-repair factors (SPL), or serving as a structural scaffold for the other modules (CORE). KAT2A/KAT2B can also associate with a second co-activator complex, the ATAC (Ada-Two-A-containing), composed of the same HAT module as SAGA (with TADA2A replacing TADA2B[25,26]), and a different CORE module, with the two complexes differing in their localisation in the cell[7] and regulating different sets of target genes[27].

Due to their key role in transcriptional activation, acetyltransferases such as KAT2A are desirable therapeutic targets, particularly in cancer[28,29]. Indeed, KAT2A has been identified as a specific vulnerability in acute myeloid leukaemia (AML)[30], while numerous studies have reported an interplay between KAT2A and MYC-driven oncogenic transcriptional programmes across different cancer subtypes[31–35]. Multiple independent genetic screens have also identified cancer vulnerabilities within non-enzymatic components of SAGA[36–40], demonstrating their functional importance and showing that targeting these components also holds therapeutic potential. Notably, inactivation of CORE and HAT components other than KAT2A often phenocopies loss of KAT2A itself[34,40], while several studies have reported a reduction in H3K9ac levels upon loss of both HAT and non-HAT SAGA components[27,34,39,41], suggesting that KAT2A's acetyltransferase activity is dependent on protein-protein interactions within SAGA. However, the contribution of individual SAGA subunits to the regulation of KAT2A protein function and stability still remains unclear.

Here, we used a fluorescence-based stability reporter to systematically measure KAT2A protein levels upon genetic perturbation of SAGA components. We found that specific components of the structural SAGA CORE module are necessary for KAT2A protein stability, recruitment to promoters and acetylation of histones. Fractionation experiments revealed loss of complex integrity and disengagement of the HAT module upon SAGA CORE disruption, resulting in the accumulation of non-complexed KAT2A protein that is prone to proteasome-mediated degradation. Using pooled CRISPR screening, we identified proteins that regulate KAT2A turnover upon perturbation of the SAGA CORE, including OTUD5 and UBR5, components of the orphan quality control system, implicating them as effectors of solitary KAT2A protein degradation. Finally, mutagenesis of a predicted UBR5 degron at the KAT2A N-terminus, which is not conserved in KAT2B, restores KAT2A abundance, highlighting a paralogue-specific degradation pathway.

## Results

### Monitoring KAT2A abundance upon knockout of SAGA components

To monitor KAT2A protein abundance upon genetic or chemical perturbation, we established a stability reporter by fusing the *KAT2A* coding sequence in-frame with BFP, followed by mCherry, with the two fluorophores separated by a P2A peptide linker. In this system, BFP therefore reports on KAT2A abundance while mCherry serves as a normalisation control, in a setup amenable to pooled or arrayed CRISPR screening using flow cytometry[38,42,43]. We expressed the KAT2A stability reporter in wild-type HAP1 cells along with Cas9 and performed arrayed CRISPR screening using single-guide RNAs (sgRNAs) targeting the 20 components of the SAGA complex (Fig. 1a, b), for which we confirmed strong and comparable editing (Supplementary Fig. 1a–d), along with positive and negative controls (Supplementary Data 1). Each sgRNA also expressed eGFP, enabling discrimination of transduced from non-transduced cells and quantification of guide RNA dropout by comparing the percentage of eGFP-positive cells for each guide at each time point of the experiment. As expected, due to their common essentiality, the eGFP signal originating from sgRNAs targeting the essential genes *RAD21* and *MYC* was rapidly lost, as was the signal from sgRNAs targeting the SAGA SPL module components *SF3B3* and *SF3B5* (Supplementary Fig. 2a–c). Thus, wild-type HAP1[Cas9] cells have high levels of Cas9 activity and allow efficient knockout of target genes, and are therefore amenable to arrayed CRISPR screening.

To validate that the KAT2A stability reporter was responsive to changes in KAT2A protein abundance, we treated HAP1[Cas9] cells with GSK-699, a potent, recently developed KAT2A/KAT2B Proteolysis Targeting Chimera (PROTAC)[34,44]. Following 6 h treatment with the PROTAC, we observed a strong reduction in KAT2A-BFP signal but no change in mCherry fluorescence (Supplementary Fig. 3a–c), as expected, as mCherry is translated as a separate polypeptide, with a >70% reduction in KAT2A-BFP fluorescence for PROTAC-treated cells compared to DMSO-treated controls (from $1.0 \pm 0.007$ (DMSO) to $0.28 \pm 0.015$ (GSK-699), mean ± s.d.). Additionally, cells transduced with sgRNAs targeting *KAT2A* itself had substantially reduced KAT2A-BFP and mCherry mean fluorescence, with a >50% reduction in signal for both fluorophores compared to *AAVS1* transduced controls (Supplementary Fig. 3d, e). Thus, the KAT2A stability reporter construct faithfully reports on KAT2A protein levels upon chemical or genetic perturbation.

To systematically investigate how loss of SAGA components affects KAT2A abundance, we gated eGFP-positive cells and measured the KAT2A-BFP/mCherry ratio for each sgRNA in our collection (Fig. 1c) relative to negative control guides targeting either the *AAVS1* safe harbour locus or non-targeting sequences. Targeting *ATXN7*, *ATXN7L3*, *USP22* and *ENY2*, components of the DUB module, did not significantly alter KAT2A levels, consistent with the observation that the DUB module is loosely bound to SAGA and hence dispensable for its structural integrity[21]. However, knockout of *TADA2B, SGF29* and *TADA3*, components of the HAT module that are in close proximity to KAT2A within SAGA, resulted in a significant drop in KAT2A abundance, indicating that KAT2A protein levels are reduced upon loss of its partners in the HAT module. Targeting *TRRAP*, the largest subunit

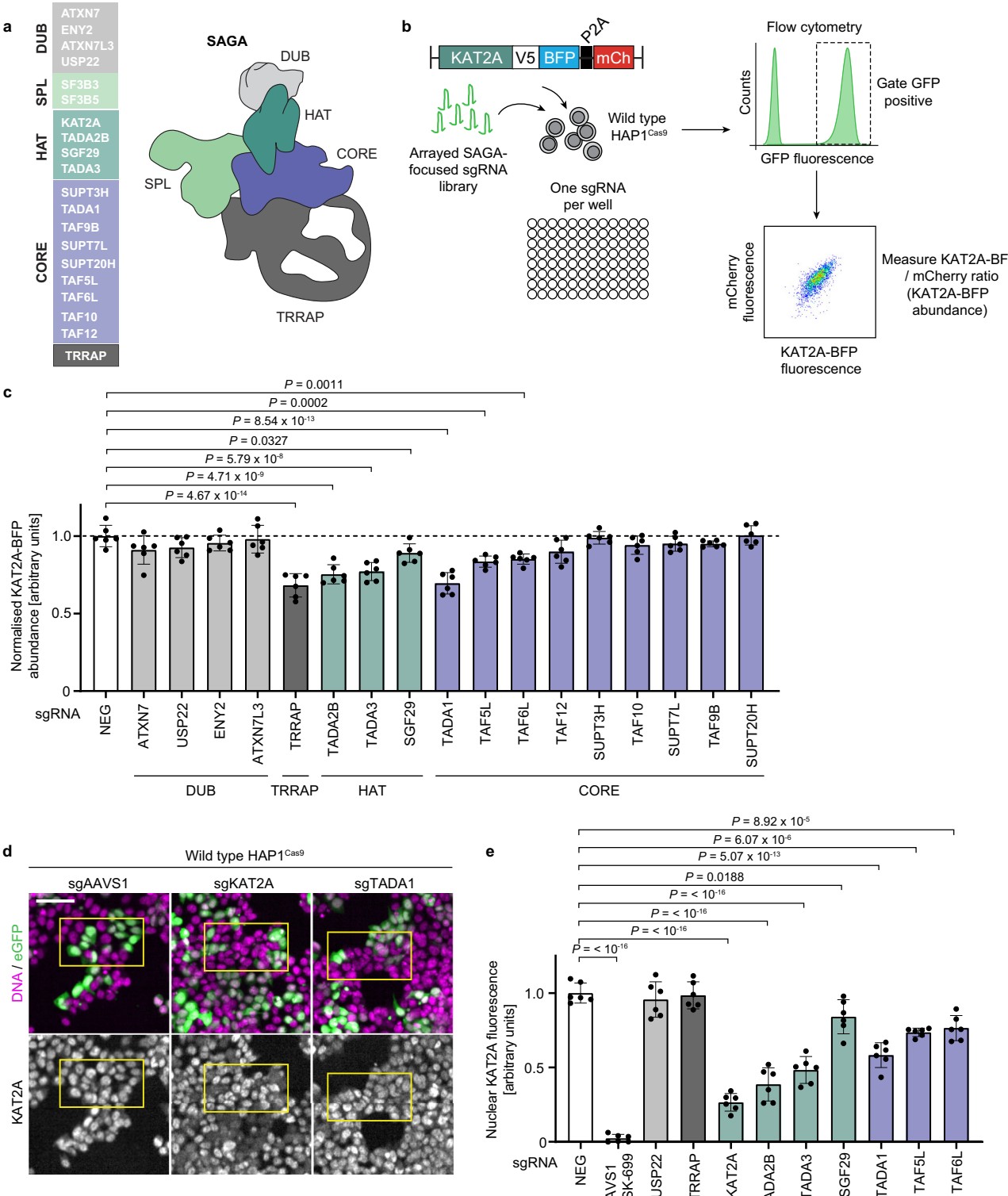

**Fig. 1 | Monitoring KAT2A abundance upon knockout of SAGA components.**
**a**, **c**, **e** SAGA CORE components are visualised in purple, histone acetyltransferase (HAT) components in dark green, splicing components (SPL) in light green, deubiquitinase (DUB) components in light grey, TRRAP in dark grey, negative controls (NEG) in white. **a** Schematic of the SAGA complex. The SAGA complex is composed of 20 proteins arranged into five modules. **b** Schematic of the KAT2A stability reporter and workflow of the arrayed CRISPR screen. **c** Quantification of KAT2A-BFP protein abundance following transduction of eGFP-expressing sgRNAs against SAGA subunits in wild-type HAP1$^{Cas9}$ cells. 5 days post-transduction, the KAT2A-BFP levels of eGFP-positive cells were quantified by flow cytometry. **d** Representative immunofluorescence images of

KAT2A and DNA/eGFP for wild-type HAP1$^{Cas9}$ cells transduced with eGFP-expressing sgRNAs against indicated genes (cells fixed 5 days post-transduction). **e** Quantification of nuclear KAT2A fluorescence by immunofluorescence for the indicated sgRNAs. Biological replicates: **c**–**e** ($n$ = 6). **c**, **e** Black dots represent the mean of each biological replicate, error bars indicate standard deviation, bars indicate the mean for each condition. Significance was tested using a one-way analysis of variance (ANOVA) with a post-hoc Dunnett's multiple comparison test. Exact $p$ values are specified in the figure. All images show single Z-slices. Yellow boxes indicate inset regions showing non-transduced (eGFP-negative) and transduced (eGFP-positive) cells. To aid visualisation, the DNA and eGFP channels are not contrast-matched. Scale bar: 50 μm.

of SAGA, also resulted in reduced KAT2A abundance using the stability reporter assay. While not structurally close to the HAT module[21], it has been reported that TRRAP and KAT2A engage in a tertiary structure with MYC[45,46], which might become disrupted upon loss of *TRRAP*. However, closer inspection of the 2D FACS plots for cells transduced with guides against *TRRAP* showed that these cells had an increase in mCherry signal compared to *AAVS1*-transduced control cells, rather than a decrease in KAT2A-BFP fluorescence (Supplementary Fig. 3f), suggesting an artefact of the reporter leads to a reduced KAT2A-BFP/mCherry ratio when targeting *TRRAP*. Interestingly, upon targeting the non-enzymatic CORE module of SAGA, we identified three subunits, *TADA1*, *TAF5L* and *TAF6L*, whose knockout also resulted in significantly reduced levels of KAT2A (Fig. 1c). These results therefore suggest that selected CORE components of SAGA, in addition to components of the HAT module, play a role in regulating KAT2A protein abundance.

In addition to SAGA, KAT2A can associate with the ATAC complex. To address whether ATAC also plays a role in regulating KAT2A abundance, we knocked out the seven unique components of ATAC (Supplementary Fig. 3g) in wild-type HAP1[Cas9] cells and measured KAT2A protein levels using the stability reporter (Supplementary Fig. 3h). While knockout of the SAGA-specific subunits *TADA1, TAF5L* and *TADA2B*, or the shared subunit *TADA3* significantly reduced KAT2A abundance, knockout of subunits unique to ATAC had little effect (Supplementary Fig. 3h). Importantly, knockout of *TADA2A* and *TADA2B*, which associate mutually exclusively with the ATAC and SAGA HAT modules respectively[25,26], had different effects on KAT2A abundance, with only loss of *TADA2B* resulting in reduced KAT2A protein levels, strongly suggesting that TADA2B (and therefore SAGA) associates with KAT2A under these conditions. Altogether, these results indicate that ATAC plays little role in regulating KAT2A abundance in HAP1 cells, which is instead primarily regulated through its association with SAGA.

To validate which components of SAGA regulate KAT2A protein abundance, we turned to high-throughput confocal microscopy and measured endogenous KAT2A levels by immunofluorescence in wild-type HAP1[Cas9] cells, following knockout of SAGA subunits that reduced KAT2A abundance in the stability reporter screen. Guides against *KAT2A* itself or treatment with GSK-699 served as positive controls, with guides against *AAVS1* or a non-targeting sequence used as negative controls. As in the arrayed screen, each sgRNA also expressed an eGFP cassette. Thus, by thresholding the nuclei of eGFP-positive cells, we could specifically measure the nuclear KAT2A fluorescence of transduced cells for each guide (Fig. 1d, e). Importantly, we observed that knockout of *KAT2A* yielded a drop in nuclear KAT2A fluorescence comparable to that induced by the PROTAC, demonstrating the sensitivity of the assay (Fig. 1d, e). Knockout of *TRRAP* had no effect on endogenous KAT2A abundance, confirming that its effect on the stability reporter in the arrayed screen was an artefact of the reporter, while loss of *USP22* also did not reduce KAT2A levels. Consistent with our screen findings, knockout of the HAT subunits *TADA2B, SGF29* and *TADA3*, as well as the CORE subunits *TADA1, TAF5L* and *TAF6L*, resulted in a significant reduction in nuclear KAT2A protein levels (Fig. 1d, e), confirming that components of the HAT module, as well as non-enzymatic structural components of SAGA, regulate KAT2A protein levels. In addition, *TAF12*, an essential gene and shared component of the SAGA CORE and TFIID complexes, which did not score as a hit in our arrayed screen, also resulted in a significant reduction in KAT2A protein levels (Supplementary Fig. 3i), equivalent to that observed upon knockout of *TADA1*. Indeed, TAF12 and TADA1 are known to fold co-translationally[7] and interact via their histone fold domains, forming a handshake[21,36], consistent with a common mechanism of action upon their knockout, leading to reduced KAT2A abundance. Importantly, knockout of other CORE subunits such as *SUPT20H, SUPT3H,*

or *TAF10* did not significantly reduce KAT2A protein levels (Supplementary Fig. 3i), demonstrating a specific effect on KAT2A abundance upon targeting *TADA1, TAF5L, TAF6L* and *TAF12*.

To further validate our findings, we turned to Cas9-expressing cells derived from two different leukaemia subtypes, the AML line MOLM-13 and the ALL (acute lymphoblastic leukaemia) line NALM6, which respectively express low and high levels of the KAT2A paralogue, KAT2B (Supplementary Fig. 4a), thereby allowing investigation of the effect of SAGA CORE perturbation in cells with different dependencies on KAT2A. As for HAP1, the KAT2A stability reporter in both cell lines was responsive to PROTAC treatment, with a substantial reduction in KAT2A-BFP abundance but no change in mCherry levels following GSK-699 treatment (Supplementary Fig. 4b, c). Importantly, knockout of *TADA1* and *TAF5L* (in both MOLM-13 and NALM6), and *TAF6L* (MOLM-13 only) also significantly reduced KAT2A-BFP abundance (Supplementary Fig. 4d, e), thus confirming a general sensitivity of KAT2A to perturbation of the SAGA CORE across multiple cell lines. Altogether, these results demonstrate that KAT2A protein levels are reduced upon loss of its binding partners in the HAT module, but also upon knockout of selected components of the non-enzymatic CORE, which do not interact with KAT2A directly but instead play a structural role in the complex, and that reduced KAT2A protein abundance upon SAGA CORE perturbation is a general feature conserved across different cellular contexts.

## Components of the SAGA CORE regulate KAT2A abundance and HAT function

To mechanistically dissect KAT2A loss upon targeting the SAGA CORE, we focused on the three SAGA-specific CORE components identified in the arrayed CRISPR screen and generated homozygous HAP1 knockout cell lines for *TAF5L, TAF6L* and *TADA1*, alongside *KAT2A* as a positive control. Clonal TAF5L, TAF6L and TADA1 KO cells showed a drastic reduction in endogenous nuclear KAT2A fluorescence, with a reduction in KAT2A fluorescence comparable to that of KAT2A KO cells (Supplementary Fig. 5a, b), with a similar reduction in nuclear KAT2A protein levels observed across multiple independent knockout clones (Supplementary Fig. 5b), confirming our previous findings with knockout pools. Importantly, overexpression of murine full-length Tada1, Taf5l or Taf6l cDNA in the corresponding KO clone restored KAT2A fluorescence to wild-type levels (Fig. 2a, b). Thus, TAF5L, TAF6L and TADA1 are necessary to maintain KAT2A protein levels.

To address whether specific domains within the SAGA CORE regulate KAT2A protein abundance, we examined publicly available cryo-EM data[21], aiming to identify candidate domains that mediate protein-protein interactions within SAGA, hypothesising that these domains might be important for the regulation of KAT2A protein levels. Based on their position in the complex bridging the HAT and CORE modules, we focused on the six WD40 repeats of TAF5L, the HEAT repeat of TAF6L, as well as an intrinsically disordered region at the TAF6L C-terminus. Indeed, overexpression of cDNA constructs lacking these domains in TAF5L (Supplementary Fig. 5c–e) or TAF6L (Supplementary Fig. 5f–h) knockout cells failed to rescue KAT2A protein levels, although a construct lacking the N-terminal domain of TAF5L rescued nuclear KAT2A fluorescence to the same extent as the full-length construct, and is therefore dispensable for regulating KAT2A protein abundance (Supplementary Fig. 5c–e). Thus, specific domains mediating protein-protein interactions within the SAGA CORE are both necessary and sufficient to maintain KAT2A protein levels.

Having established that loss of *TAF5L, TAF6L*, or *TADA1* reduces KAT2A protein levels, we wished to address if their loss also resulted in global changes in H3K9 acetylation, a mark associated with active promoters, and the main histone mark deposited by KAT2A on chromatin. To quantitatively address this question, we measured H3K9ac fluorescence by confocal microscopy in HAP1 knockout pools 6 days after sgRNA transduction, with eGFP again used to identify transduced

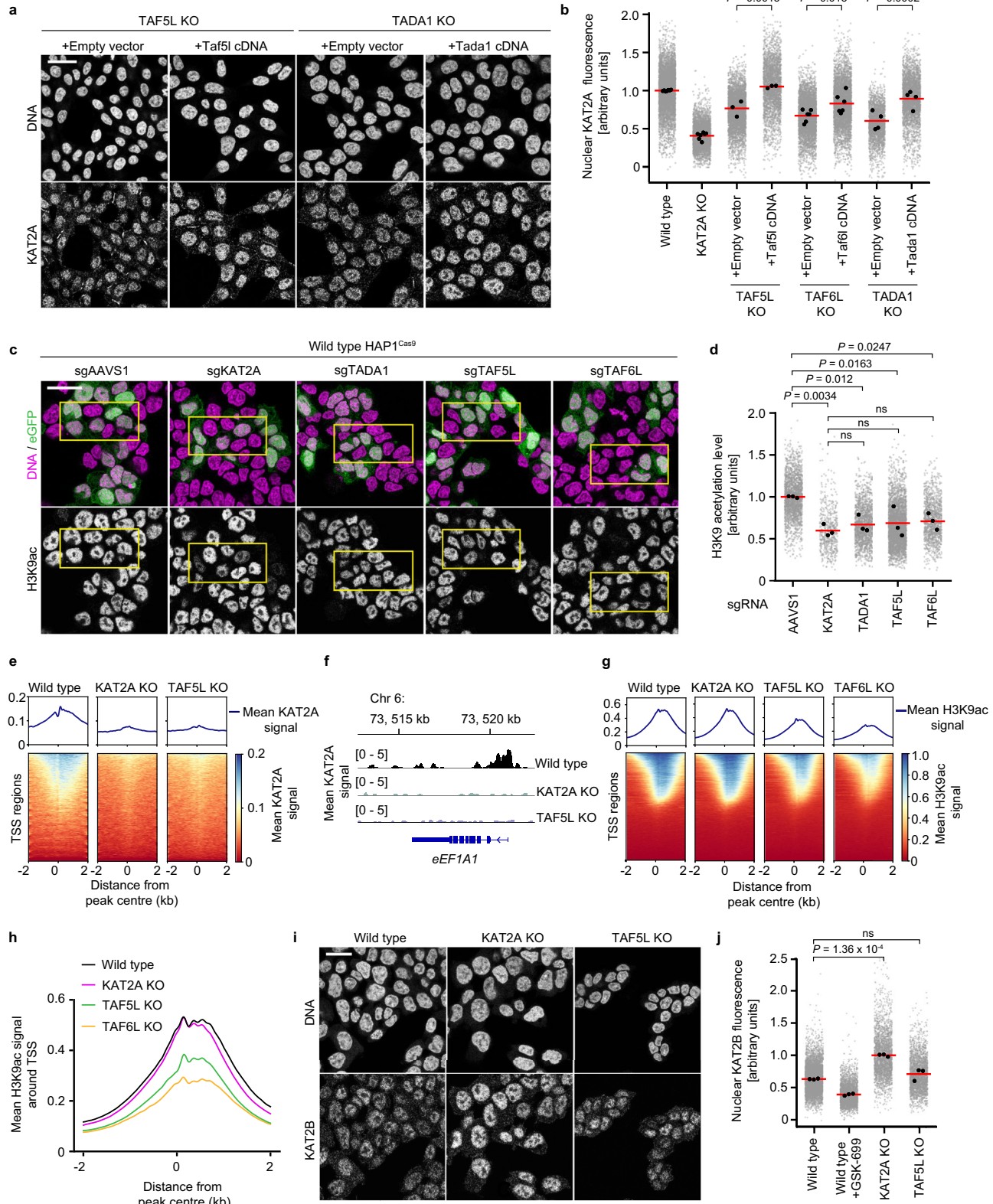

cells. Indeed, knockout of *TADA1*, *TAF5L* or *TAF6L* led to a substantial reduction in global H3K9ac levels compared to *AAVS1*-transduced controls (Fig. 2c, d), consistent with previous findings in mouse embryonic stem cells[41]. Interestingly, the three sgRNAs targeting SAGA CORE components resulted in a global reduction in H3K9 acetylation comparable to that observed upon targeting *KAT2A* (relative H3K9ac signal: sgKAT2A: 0.60 ± 0.07, sgTAF5L: 0.69 ± 0.18, sgTAF6L:

0.70 ± 0.10, sgTADA1: 0.67 ± 0.10, mean ± s.d.). Thus, despite residual amounts of KAT2A protein in the nucleus in CORE KO cells (Fig. 2b, Supplementary Fig. 5b), KAT2A's histone acetylation output is reduced globally upon loss of *TAF5L, TAF6L, or TADA1*.

As perturbing the SAGA CORE phenocopies loss of HAT activity at the level of global H3K9 acetylation, we hypothesised that recruitment of KAT2A to chromatin is abrogated in SAGA CORE mutants. To test

**Fig. 2 | Components of the SAGA CORE regulate KAT2A protein abundance and HAT function. a** Representative immunofluorescence images of KAT2A and DNA for TAF5L KO or TADA1 KO HAP1 cells stably overexpressing an empty vector or murine cDNA. **b** Quantification of nuclear KAT2A fluorescence by immunofluorescence for conditions as indicated. **c** Representative immunofluorescence images of H3K9ac and DNA/eGFP for wild-type HAP1^Cas9 cells transduced with the indicated eGFP-expressing sgRNAs (cells fixed 6 days post-transduction). **d** Quantification of H3K9 acetylation by immunofluorescence for conditions as indicated. **e** Pile up of KAT2A signal in a 4 kb window centred around TSS as determined by CUT&RUN. **f** Genomic tracks of KAT2A signal at the *eEF1A1* locus. **g** Pile up of H3K9ac signal in a 4 kb window centred around TSS as determined by CUT&RUN. **h** Line profiles of mean H3K9ac signal around TSS for conditions as indicated. **i** Representative immunofluorescence images of KAT2B and DNA for wild-type, KAT2A KO, or TAF5L KO HAP1 cells. **j** Quantification of nuclear KAT2B fluorescence by immunofluorescence for conditions as indicated. Biological replicates: **a, b** (TAF5L KO ($n = 3$), TAF6L KO ($n = 6$), wild type ($n = 7$), KAT2A KO ($n = 7$), TADA1 KO ($n = 4$); **c, d, i, j** ($n = 3$); **e–h** ($n = 2$). **b, d, j** Grey dots represent the mean of individual nuclei, black dots represent the mean of each biological replicate and red bars indicate the mean of biological replicates. Significance was tested using a one-way ANOVA with a post-hoc Dunnett's multiple comparison test. Exact *p* values are specified in the figure. All images show single Z-slices. Yellow boxes indicate inset regions showing non-transduced (eGFP-negative) and transduced (eGFP-positive) cells. To aid visualisation, the DNA (except Fig. 2d) and eGFP channels are not contrast-matched. Scale bars: 20 μm.

this hypothesis, we measured KAT2A chromatin occupancy by CUT&RUN[47]. While we could detect enrichment of KAT2A signal around transcriptional start sites (TSSs) in wild-type cells, knockout of *TAF5L* phenocopied loss of *KAT2A*, as we were unable to detect KAT2A on chromatin in either KAT2A or TAF5L KO cells (Fig. 2e, f). In line with loss of KAT2A binding, we also observed a reduction in H3K9ac signal around TSS regions in TAF5L KO and TAF6L KO clones (Fig. 2g, h), demonstrating that perturbation of the structural SAGA CORE not only leads to loss of KAT2A recruitment to promoters, but that histone acetylation is also reduced at these regions. However, the H3K9ac signal in KAT2A KO cells was largely unchanged compared to wild type (Fig. 2g, h), suggesting that compensatory mechanisms exist in KAT2A KO cells that can largely retain H3K9ac at promoters. Indeed, prior work has shown that upon KAT2A loss, its paralogue KAT2B is upregulated in neuroblastoma[34], and that loss of both KAT2A and KAT2B is necessary to reduce H3K9ac levels[34,40,48]. Consistent with these findings, although KAT2B is normally only weakly expressed in HAP1 cells (Supplementary Fig. 6a), KAT2B protein levels were strongly upregulated in KAT2A KO cells, although KAT2B transcript levels were not significantly increased (Supplementary Fig. 6b–f), suggesting that in the long-term absence of KAT2A, KAT2B can, to a large extent, compensate for its absence. Importantly, knockout of *TAF5L* did not lead to upregulation of KAT2B protein levels (Fig. 2i, j), indicating that, unlike targeting *KAT2A* itself, disruption of SAGA CORE components leads to loss of KAT2A from chromatin without triggering compensatory pathways that upregulate KAT2B. Thus, not only is KAT2A protein abundance reduced upon perturbing the SAGA CORE, but the residual protein cannot be recruited to its canonical binding sites on chromatin around promoters, resulting in decreased H3K9ac deposition.

## Loss of TAF5L and TADA1 leads to HAT module disengagement from SAGA

SAGA CORE perturbation has previously been reported to lead to impaired SAGA complex assembly[27,49]. However, how the HAT module interacts with the CORE and if an intact CORE is required for engagement of the HAT with SAGA is not well characterised. As we did not detect chromatin-bound KAT2A at promoters upon targeting the SAGA CORE, we reasoned that TAF5L, TAF6L and TADA1 might mediate important protein-protein interactions within the CORE that are necessary for engagement of the HAT module with SAGA. In such a situation, loss of these proteins would lead to the separation of the HAT module and an increase in non-complexed KAT2A protein that can no longer bind to chromatin. To test this hypothesis, we turned to sucrose gradient fractionation (Fig. 3a). High-molecular weight species, such as the intact SAGA complex, containing all or most of its components, have high density and thus migrate into the dense sucrose fractions, while solitary proteins or individual complex modules of low-molecular weight remain near the top of the gradient (Fig. 3a).

To investigate to what extent perturbation of the SAGA CORE affects engagement of the HAT module with SAGA, we ultracentrifuged and fractionated sucrose gradients from wild-type, TAF5L KO and TADA1 KO cells, before western blotting against KAT2A, as well as

TADA3, another HAT component. In wild-type cells, KAT2A and TADA3 could be detected across the full range of fractions (Fig. 3b, c), indicating the presence of both complexed and non-complexed forms of the proteins. Indeed, in wild-type cells, both KAT2A and TADA3 were readily detectable in high-molecular weight fractions (heavy fractions) (Fig. 3b, c, red boxes), consistent with their incorporation into fully assembled SAGA complexes, where the HAT module is engaged with the rest of the complex. Strikingly, in the absence of TAF5L and TADA1, we observed a pronounced loss of KAT2A and TADA3 signal in heavy fractions (Fig. 3b, c, red boxes), consistent with loss of complex integrity and disengagement of the HAT from the rest of SAGA. Importantly, overexpression of murine Tada1 cDNA in TADA1 KO cells restored both KAT2A and TADA3 to heavy fractions (Supplementary Fig. 7a, b, red boxes), largely recapitulating their distribution in wild-type cells, indicating that the reestablishment of an intact SAGA CORE is sufficient to restore the engagement and retention of the HAT module with SAGA. Altogether, these results therefore demonstrate that TAF5L and TADA1 are not only required to maintain KAT2A protein levels, but are also important regulators of SAGA complex integrity that are necessary to maintain engagement of the HAT module with SAGA.

As perturbation of the SAGA CORE results in loss of complex integrity and dissociation of HAT components from the complex, we asked if KAT2A might instead associate with ATAC when the SAGA CORE is disrupted. We hypothesised that in CORE KO cells, where the HAT module dissociates from SAGA, if KAT2A is able to associate with ATAC, then KAT2A levels would be reduced upon knockout of ATAC components, while KAT2A abundance should remain unchanged if KAT2A and ATAC do not associate with each other. To test this, we knocked out ATAC components as well as positive and negative controls in TAF5L KO HAP1^Cas9 cells expressing the KAT2A stability reporter, and measured KAT2A-BFP abundance by flow cytometry (Supplementary Fig. 7c). In contrast to our findings in wild-type cells, where loss of ATAC subunits had little effect on KAT2A abundance (Supplementary Fig. 3h), knockout of numerous ATAC components significantly reduced KAT2A abundance in TAF5L KO cells, including the structural components *ZZZ3, YEATS2* and *DR1*, as well as the HAT subunit *TADA2A* (Supplementary Fig. 7c), consistent with ATAC components contributing to the regulation of KAT2A protein levels when the SAGA complex is disrupted. In particular, as binding of *TADA2A* and *TADA2B* to the HAT module is mutually exclusive, reduced KAT2A abundance upon *TADA2A* loss in TAF5L KO cells (Supplementary Fig. 7c) but not in wild type (Supplementary Fig. 3h) suggests that KAT2A associates with ATAC specifically when SAGA is not available. Altogether, these results support that although KAT2A primarily associates with SAGA in HAP1 cells, a proportion of the KAT2A pool can associate with ATAC when assembled SAGA is no longer available following disruption of its structural CORE.

## Progressive destabilisation of CORE and HAT upon depletion of TADA1

Constitutive knockout of *TAF5L, TAF6L* and *TADA1* leads to reduced KAT2A protein abundance, but as the rate of protein rundown

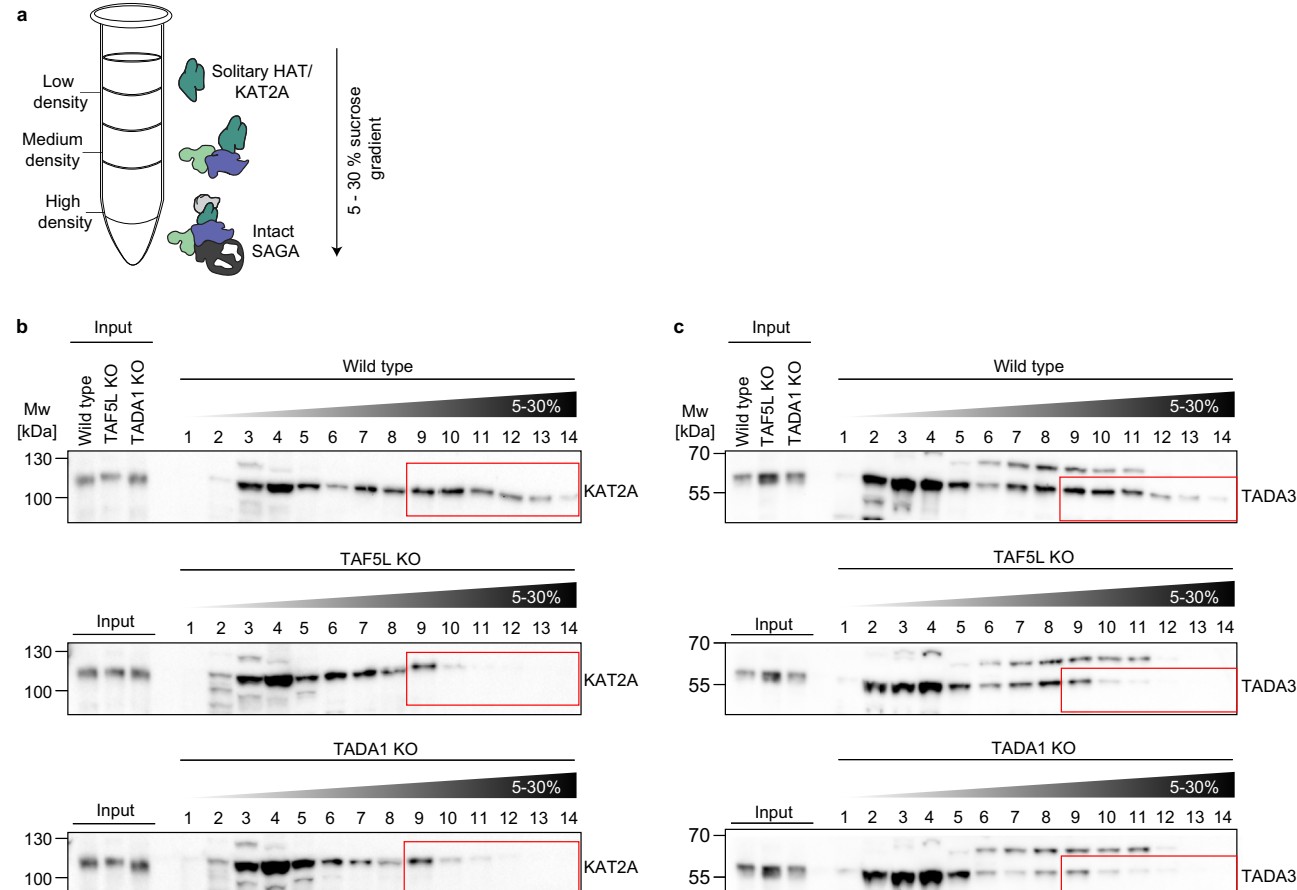

**Fig. 3 | Loss of TAF5L and TADA1 leads to HAT module disengagement from SAGA. a** Schematic of the expected distribution of SAGA complex components following sucrose gradient ultracentrifugation. High-molecular-weight species sediment into the denser sucrose while low-molecular-weight species remain near the top of the gradient. **b, c** Sucrose gradient fractionation reveals that the SAGA HAT disengages from SAGA when the CORE is perturbed. Cell lysates were prepared from wild-type, TAF5L KO and TADA1 KO cells, and loaded onto 5–30% sucrose gradients. Gradients were ultracentrifuged and fractionated to separate protein complexes based on their sedimentation. The resultant fractions were loaded onto an SDS-PAGE gel and analysed by western blotting, using antibodies

against KAT2A (**b**) or TADA3 (**c**) as indicated. Membranes were developed simultaneously to allow cross-comparison between genotypes. Inputs indicate the cell lysates for each condition prior to ultracentrifugation. Numbers indicate each fraction, moving from lowest to highest density sucrose fractions (5–30%). Red boxes indicate fractions containing high-molecular-weight KAT2A or TADA3 species that are retained in wild-type cells and absent in TAF5L KO or TADA1 KO cells. **b** Immunoblot analysis of KAT2A following sucrose gradient fractionation for conditions as indicated. **c** Immunoblot analysis of TADA3 following sucrose gradient fractionation for conditions as indicated. Biological replicates: **b, c** ($n = 3$).

following gene knockout cannot be temporally controlled, in such a setup, the kinetics of protein loss cannot be addressed. Therefore, to assess the kinetics of KAT2A protein depletion upon perturbing the SAGA CORE, we expressed a TADA1-FKBP12$^{F36V}$ (dTAG) degron construct[50] in TADA1 KO HAP1 cells, using a haemagglutinin (HA) tag to monitor TADA1 protein levels, with overexpression of *Tada1* restoring nuclear KAT2A levels to those of wild type (Fig. 2b). Immunostaining and confocal microscopy revealed that although TADA1 protein was rapidly degraded within 1 h of dTAG$^{V}$-1 addition (Fig. 4a, b), KAT2A protein levels gradually reduced over time, reaching close to the level of TADA1 KO cells after 24 h of dTAG$^{V}$-1 treatment (Fig. 4c, d). Thus, acute degradation of TADA1 further confirms that KAT2A protein abundance is reduced when the SAGA CORE is perturbed. However, the effect of TADA1 loss on KAT2A is not immediate, with KAT2A protein levels instead decreasing progressively over time.

Having observed that KAT2A protein levels are reduced following acute degradation of TADA1, we wished to address the consequences of SAGA CORE perturbation on other components of SAGA, as well as the effect globally on other protein complexes. We therefore used TMT-labelling to perform expression proteomics for 5 time points of interest in triplicate, following TADA1 degradation for 1, 6, 24, or 48 h,

and compared protein abundancies relative to control cells treated with DMSO. More than 7200 proteins were successfully identified in our dataset, including 15 of the 20 components of SAGA (Fig. 4e, Supplementary Fig. 8a). Consistent with our immunofluorescence data (Fig. 4a, b), TADA1 was significantly depleted after 1 h of dTAG$^{V}$-1 treatment and remained the most strongly depleted protein at all time points (Supplementary Fig. 8b–e). In the absence of unique KAT2A peptides, we annotated common KAT2A/KAT2B peptides to estimate KAT2A abundance and again observed progressive depletion over time (Fig. 4e), with the most prominent depletion after 48 h of dTAG$^{V}$-1 treatment. Importantly, although it was not possible to identify unique KAT2A peptides due to its high sequence similarity with KAT2B, the low expression of KAT2B in HAP1 cells (Supplementary Fig. 6a, e, f) and lack of compensatory KAT2B upregulation upon CORE perturbation (Fig. 2i, j) suggests that the majority of annotated KAT2A/KAT2B peptides in our dataset do indeed correspond to KAT2A. In addition to KAT2A, the abundancies of several other SAGA components also progressively decreased over time, most prominently the CORE components TAF5L and TAF12, as well as TRRAP, with TAF6L and the HAT component TADA3 also significantly depleted after 48 h dTAG$^{V}$-1 treatment (Fig. 4e, Supplementary Fig. 8d, e). The abundance of

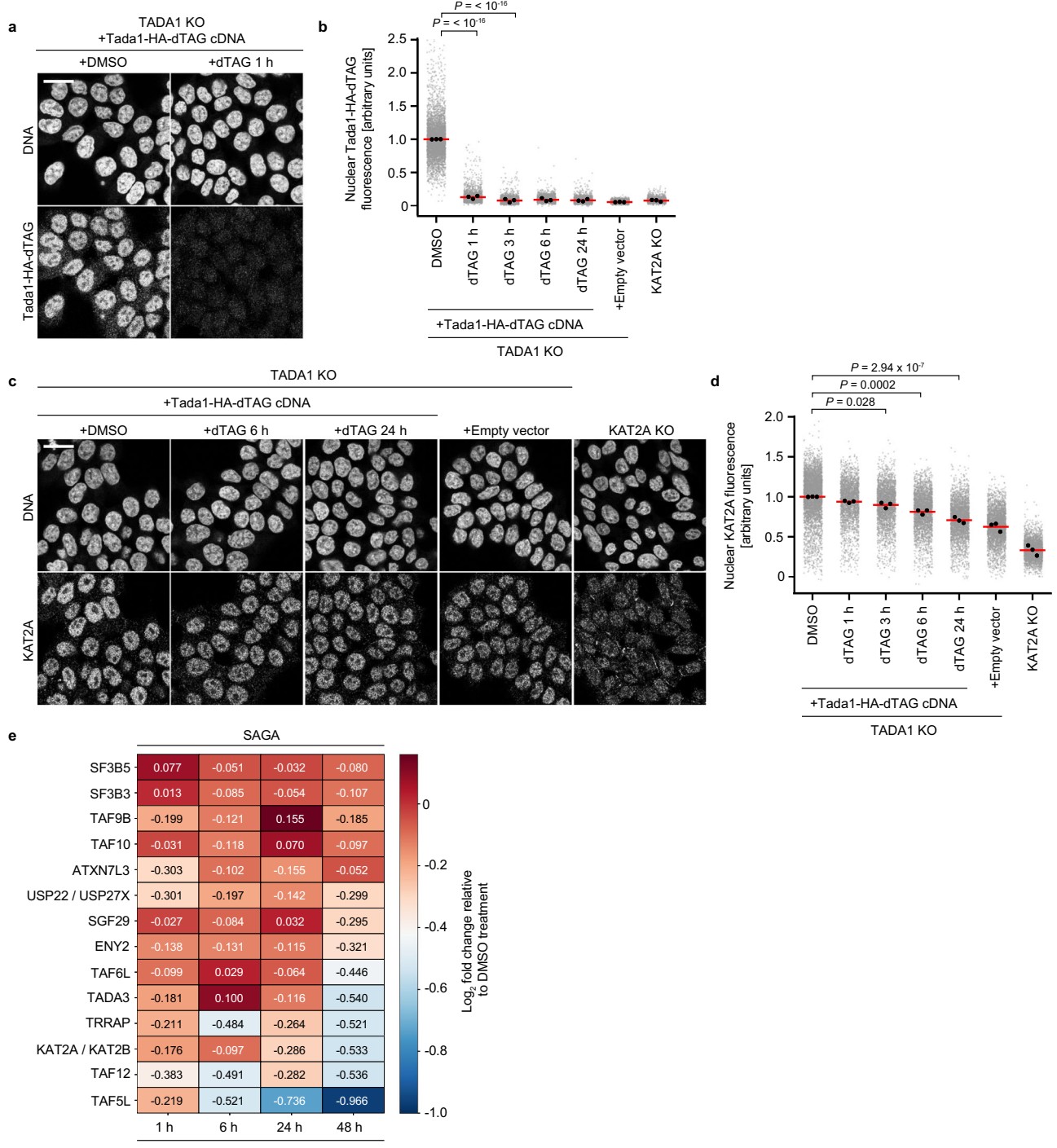

**Fig. 4 | Progressive destabilisation of SAGA CORE and HAT upon depletion of TADA1. a** Representative immunofluorescence images of TADA1-HA and DNA for TADA1 KO HAP1 cells stably overexpressing full-length TADA1-HA-dTAG cDNA. Cells were treated with DMSO or 100 nM dTAG$^V$-1 for time points as indicated. **b** Quantification of nuclear TADA1-HA fluorescence by immuno-fluorescence for cells treated with compounds as indicated. **c** Representative immunofluorescence images of KAT2A and DNA for TADA1 KO cells stably overexpressing full-length TADA1-HA-dTAG cDNA, treated with DMSO or dTAG$^V$-1 for time points as indicated. **d** Quantification of nuclear KAT2A fluorescence by immunofluorescence for cells treated with compounds as

indicated. **e** Heatmap of differential expression of SAGA complex components following acute depletion of TADA1 for time points as indicated, as determined by TMT-expression proteomics. Log$_2$ fold changes of SAGA subunits relative to DMSO-treated controls are shown for each subunit at each time point. **b**, **d** Biological replicates: **a**–**e** ($n$ = 3). **b**, **d** Grey dots represent the mean of individual nuclei, black dots indicate the mean of each biological replicate and red bars indicate the mean of biological replicates. Significance was tested using a one-way ANOVA with a post-hoc Dunnett's multiple comparison test. Exact $p$ values are specified in the figure. All images show single Z-slices. To aid visualisation, the DNA channel is not contrast-matched. Scale bars: 20 μm.

components from other protein complexes that share subunits with SAGA was, however, largely unchanged, with minimal changes in the levels of ATAC and TFIID subunits (Supplementary Fig. 8b–e). Thus, acute depletion of TADA1 not only leads to reduced KAT2A protein levels but also to the progressive loss of numerous other components of SAGA upon prolonged TADA1 depletion.

Interestingly, several of the downregulated components are known to directly interact with or are in close structural proximity to TADA1 within SAGA. Indeed, TAF12 interacts with TADA1 via its histone fold domain[36], in a process that occurs co-translationally[7]. Similarly, TRRAP has been suggested to form interactions with TADA1 via a cleft bridging TRRAP and the CORE[21], with TAF5L and TAF6L also in close proximity to TADA1[21]. As co-translational assembly is a prominent mechanism by which large multi-protein complexes assemble[3,4,51], loss of direct interactors or neighbouring proteins can lead to destabilisation of other complex members[17]. Consistent with this finding, we observed loss of both TAF12 and TAF5L at early time points after TADA1 depletion, indicating that these components are particularly sensitive to the loss of TADA1-mediated protein-protein interactions, and that acute degradation of TADA1 might lead to collateral degradation of TAF12 due to the loss of its normal co-translational assembly pathway. Although KAT2A is not known to directly interact with TADA1, its abundance and engagement with SAGA depends on an intact structural CORE (Figs. 1–3), which becomes increasingly destabilised over time due to loss not only of TADA1, but the additional CORE subunits TAF12, TAF5L and TAF6L (Fig. 4e). These results are therefore consistent with a cumulative disruption of the CORE-HAT interface following TADA1 depletion, which leads to HAT module disengagement and an increase in non-complexed KAT2A that becomes progressively destabilised in the absence of its normal binding partners. Altogether, these results indicate that acute perturbation of the SAGA CORE via depletion of TADA1 results in collateral loss of selected SAGA components, particularly for proteins in the CORE and HAT modules that are structurally proximal to or direct interactors of TADA1.

## Non-complexed KAT2A is degraded in a process mediated by UBR5 and OTUD5

Perturbation of the SAGA CORE reduces KAT2A protein abundance and also leads to HAT disengagement from the complex. We therefore hypothesised that in CORE KO cells, where complex integrity is perturbed, regions of KAT2A that in the intact SAGA complex are buried or bound by interaction partners might become exposed, resulting in the formation of degrons that target KAT2A to the proteasome and lead to its degradation. To assess if KAT2A protein loss in CORE KO mutants is indeed proteasome-sensitive, we treated TAF5L KO HAP1 cells with inhibitors of key components of the ubiquitin-proteasome system (UPS), including neddylation enzymes (MLN-4924), ubiquitin E1-activating enzymes (TAK-243), and the proteasome itself (MG-132), and measured endogenous KAT2A levels by immunofluorescence. Treatment with all three inhibitors rescued KAT2A protein levels in TAF5L KO cells (Fig. 5a, b), with the strongest rescue observed upon inhibition of ubiquitin E1-enzyme activation or the proteasome, with similar results observed in TAF6L and TADA1 KO cells (Supplementary Fig. 9a), indicating a general proteasome-sensitivity of KAT2A when the SAGA CORE is perturbed. Crucially, in the absence of inhibitors, KAT2A mRNA levels remained unchanged or even increased in CORE KO cells compared to wild type (Supplementary Fig. 9b, c), demonstrating that loss of KAT2A protein is not due to reduced transcription. Thus, loss of KAT2A protein in the absence of TAF5L, TAF6L, or TADA1 is the result of proteasome-mediated degradation.

Having determined that the loss of KAT2A protein is proteasome-sensitive, we wished to understand the molecular mechanism of

KAT2A degradation in CORE KO mutants and identify which effectors of the UPS, including the >600 E3 ligases in the human genome, target KAT2A for degradation. To this end, we again made use of the KAT2A stability reporter (Fig. 1). Treatment of TAF5L KO or TAF6L KO cells expressing the stability reporter with inhibitors of the proteasome machinery also rescued KAT2A-BFP protein levels (Fig. 5c, Supplementary Fig. 9d), enabling screening for effectors of KAT2A degradation. To identify specific effectors regulating KAT2A protein stability, we therefore performed a FACS-based pooled CRISPR screen in TAF5L KO HAP1[Cas9] cells with a library targeting 1301 Ubiquitin-associated human genes, with 6 sgRNAs per gene[52] (Fig. 5d). By sorting cells expressing high levels of KAT2A-BFP (KAT2A-BFP[high]) after library transduction (Supplementary Fig. 9e) and performing next-generation sequencing (NGS), we could identify genes whose knockout promotes KAT2A stabilisation to nominate candidate regulators of KAT2A degradation when the SAGA CORE is perturbed.

Interestingly, knockout of USP22, which confers the enzymatic activity of the SAGA DUB module and could, in principle, auto-deubiquitinate KAT2A, did not alter KAT2A abundance (Fig. 5e), thereby ruling out that interactions between the HAT and DUB modules regulate KAT2A stability. Consistent with our findings with inhibitors, loss of components of the neddylation machinery, which are required for activation of Cullin Ring E3 Ligases (CRLs), scored as top hits in the screen, including the E1-enzymes NAE1 and UBA3, which form a heterodimer, the E2-conjugating enzyme UBE2M, and NEDD8, which encodes the neddylation modification itself (Fig. 5e). However, despite this neddylation sensitivity, we found only weak enrichment of CRL components, such as the E3 ligase substrate receptor VHL and the E3 scaffolding protein CUL3, suggesting non-specific or promiscuous degradation of KAT2A by CRLs that is abrogated only upon inhibition of all of them, which would occur upon loss of neddylation pathway components (Fig. 5e). We however uncovered numerous candidate non-CRL E3 ligases that were significantly enriched in KAT2A-BFP[high] cells, such as the DNA-repair associated TRAIP[53], the RBR (RING-between-RING) E3 ligase ARIH1, and UHRF1[54], a known chromatin-associated factor. We also identified the HECT-type E3 ligase UBR5[55–57], in addition to its interactor, the deubiquitinase OTUD5[58], as significant hits whose loss stabilised KAT2A in the absence of TAF5L (Fig. 5e).

To validate which of the candidate UPS effectors promote KAT2A degradation upon SAGA CORE perturbation, we individually targeted the top hits from the pooled screen using the two highest scoring sgRNAs per gene in TAF5L KO HAP1[Cas9] cells, and used confocal microscopy to measure endogenous KAT2A protein levels, again using eGFP to identify transduced cells. To better assess the relative rescue of KAT2A protein levels in CORE KO cells, wild-type HAP1[Cas9] cells were transduced with the same guides, with cells treated with GSK-699 or MG-132 serving as controls. Consistent with the pooled CRISPR screen, knockout of the neddylation pathway components UBE2M and UBA3 rescued KAT2A protein levels in TAF5L KO cells (Fig. 5f, g), although only partially, reaching approximately 50% of the levels observed upon proteasome inhibition. Loss of neddylation components was, however, as effective as proteasome inhibition in wild-type cells, with a mild increase in nuclear KAT2A signal (Supplementary Fig. 9f, g), indicative of a reduced contribution of neddylation-dependent E3 ligases to KAT2A protein degradation when the SAGA CORE is perturbed.

Interestingly, the magnitude of rescue of KAT2A protein levels upon proteasome inhibition was substantially higher in TAF5L KO cells ($1.41 \pm 0.13$, mean ± s.d.) compared to wild type ($1.16 \pm 0.02$) (Fig. 5f, g, Supplementary Fig. 9f, g). Similar results were also observed in TADA1 KO cells (Supplementary Fig. 9h, i), indicating a general increase in the sensitivity of KAT2A to proteasome-mediated degradation when the CORE is perturbed. Such findings are consistent with loss of complex integrity in CORE KO cells (Fig. 3b, c), where a greater proportion of KAT2A likely exists outside of the intact SAGA complex, away from its normal binding partners. Loss of binding interactions with other SAGA

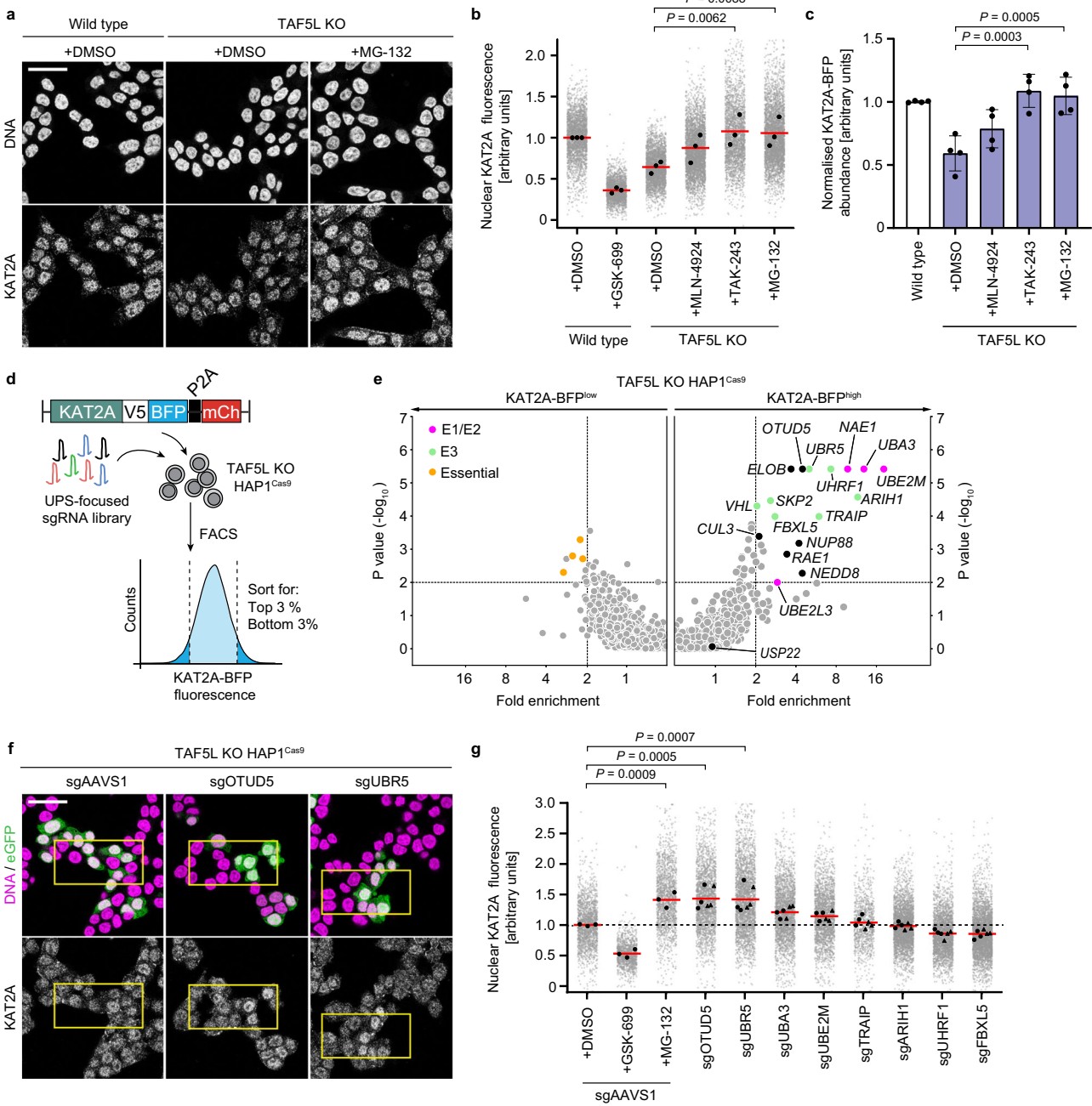

**Fig. 5 | Non-complexed KAT2A is degraded in a process mediated by UBR5 and OTUD5. a** Representative immunofluorescence images of KAT2A and DNA for wild-type and TAF5L KO cells treated with compounds as indicated. **b** Quantification of nuclear KAT2A fluorescence by immunofluorescence following compound treatment for the indicated conditions. **c** Quantification of KAT2A-BFP abundance by flow cytometry for wild-type (white bars) and TAF5L KO (purple bars) cells following treatment with indicated compounds. **d** Schematic of FACS-based CRISPR screen in TAF5L KO[Cas9] cells. Cells were transduced with a UPS (ubiquitin-proteasome focused)-focused sgRNA library and sorted based on KAT2A-BFP levels. **e** KAT2A stability reporter screen in TAF5L KO cells. Fold changes and *p* values of the KAT2A-BFP[high] and KAT2A-BFP[low] populations were calculated compared to the KAT2A-BFP[mid] population using a two-sided negative binomial test (MAGeCK). Significant hits: Fold-enrichment ≥2 and −log[10] *p* values ≥ 2. E1/E2 enzymes (magenta dots), E3 substrate receptors (green dots), and essential genes (orange dots) are highlighted. **f** Representative immunofluorescence images of

KAT2A and DNA/eGFP for TAF5L KO HAP1[Cas9] cells transduced with eGFP-expressing sgRNAs as indicated (cells fixed 5 days post-transduction). **g** Quantification of nuclear KAT2A fluorescence by immunofluorescence for TAF5L KO HAP1[Cas9] cells transduced with the indicated sgRNAs. Biological replicates: **a**, **b**, **e**–**g** (*n* = 3); **c** (*n* = 4). Technical replicates: **c** (*n* = 3). **b**, **g** Grey dots represent the mean of individual nuclei. **b**, **c** Black dots represent the mean of each biological replicate. **b**, **g** Red bars indicate the mean of biological replicates. **c** Error bars indicate standard deviation; bars indicate the mean for each condition. **g** Two independent sgRNAs are merged for UPS genes. Black dots or triangles represent the mean for each guide. Significance was tested using a one-way ANOVA with a post-hoc Dunnett's multiple comparison test. Exact *p* values are specified in the figure. All images show single Z-slices. Yellow boxes indicate inset regions showing non-transduced (eGFP-negative) and transduced (eGFP-positive) cells. To aid visualisation, the DNA and eGFP channels are not contrast-matched. Scale bars: 20 μm.

components upon SAGA CORE disruption might therefore render KAT2A more susceptible to collateral protein degradation when complex integrity is disrupted, and thus more responsive to stabilisation upon proteasome inhibition compared to wild-type cells, where the HAT module remains engaged with SAGA.

Most strikingly, knockout of the HECT-type E3 ligase *UBR5* and its interactor *OTUD5* rescued nuclear KAT2A protein levels to the same extent as proteasome inhibition in both TAF5L KO and TADA1 KO cells (Fig. 5f, g, Supplementary Fig. 9h, i). In contrast, although knockout of *OTUD5* and *UBR5* also increased KAT2A levels in wild-type cells, the magnitude of rescue was substantially lower compared to CORE KO cells (Supplementary Fig. 9f, g), implicating UBR5 and OTUD5 as important regulators of KAT2A degradation specifically when the SAGA CORE is perturbed. Importantly, knockout of both *OTUD5* and *UBR5* in two different AML cell lines also increased KAT2A protein abundance compared to *AAVS1*-transduced controls (Supplementary Fig. 9j), indicating that OTUD5 and UBR5 are common regulators of KAT2A protein abundance across different cell lines, including in disease-relevant models. Consistent with UBR5-mediated KAT2A degradation, OTUD5 is known to stabilise UBR5 via lysine deubiquitination, with loss of OTUD5 thus also reducing UBR5 protein levels[58]. Intriguingly, UBR5 was recently identified as a key regulator of homeostatic protein turnover, functioning as an E3 ligase dedicated to the degradation of solitary or orphan proteins, including transcription factors such as MYC, and nuclear hormone receptors[13,59,60]. Our results, therefore, identify UBR5 as an important regulator of KAT2A protein levels upon SAGA CORE disruption and are consistent with the recognition of non-complexed KAT2A by UBR5 as an orphan substrate when the SAGA CORE is perturbed, leading to its collateral degradation.

### Paralogue-specific residues destabilise KAT2A

Given the high sequence homology between KAT2A and its paralogue KAT2B, we wished to understand if KAT2B also becomes collaterally degraded upon disruption of the SAGA CORE. The expression of KAT2B is, however, very low in wild-type HAP1 cells (Supplementary Fig. 6a, f), precluding investigation of the effect of CORE perturbation in a wild-type background. As KAT2B protein expression is strongly upregulated in KAT2A KO cells (Fig. 2i, j, Supplementary Fig. 6f), we therefore knocked out the CORE components *TAF5L, TAF6L* and *TADA1* in KAT2A KO cells, and measured endogenous KAT2B fluorescence compared to *AAVS1*-transduced controls by confocal microscopy (Fig. 6a, b). Interestingly, KAT2B protein levels were unchanged following SAGA CORE disruption (Fig. 6b), indicating that, unlike KAT2A, KAT2B does not become destabilised. Thus, although KAT2A undergoes collateral degradation following SAGA CORE disruption in a process mediated by OTUD5 and UBR5, KAT2B protein levels are unaffected upon CORE perturbation, suggesting a paralogue-specific degradation pathway that acts upon KAT2A but not KAT2B.

To address which residues of KAT2A might mediate its destabilisation, we considered the known degron substrates of UBR5. UBR5 does not recognise a single canonical degron motif, although in vitro it is able to recognise type-I N-degrons with positively charged amino acids at their N-terminus via its UBR-box domain[61]. Consistent with its role in recognising orphan and non-complexed protein subunits[13,60], UBR5 has, however, been shown to recognise a number of different degron motifs, typically enriched in hydrophobic or basic residues that become exposed following loss of binding to complex partners, with specificity of substrate recognition achieved via sandwiching of the substrate between tetrameric UBR5 assemblies[13,56,57,60].

The differential sensitivity of KAT2A and KAT2B to SAGA CORE disruption, however, suggests that the degron targeting KAT2A for degradation might be absent in KAT2B. Inspection of the protein sequences of KAT2A and KAT2B revealed that although the two proteins share greater than 70% sequence homology, they differ in their

N-termini (Fig. 6c). Focusing on this region of KAT2A, we identified a short motif (RPLGS) uniquely found at the KAT2A N-terminus as a candidate UBR5 degron motif and cloned a KAT2A stability reporter construct (KAT2A$^{N-term}$) where both the arginine and leucine residues were mutated to alanine (Fig. 6d). In addition, we cloned mutant reporter constructs targeting lysines immediately downstream of KAT2A$^{N-term}$, a leucine rich region found in both KAT2A and KAT2B, and a stretch of five lysines at the C-terminus of both proteins, in the proximity of an RQK motif previously identified as a UBR5 degron in MCRS1[13]. To test these putative motifs and their effect on KAT2A protein levels, we expressed the four constructs in TAF5L KO HAP1 cells and measured KAT2A-BFP abundance by flow cytometry (Fig. 6e). Strikingly, and in agreement with our data showing that KAT2A but not KAT2B is collaterally degraded, KAT2A levels were elevated only in cells expressing the KAT2A$^{Nterm}$ mutant (Fig. 6e), implicating KAT2A-specific residues at its N-terminus as important regulators of its stability, while also suggesting that the lysines immediately downstream of the degron are not the sites of ubiquitination. Altogether, these results therefore identify a minimal, paralogue-specific degron at the KAT2A N-terminus that governs its stability following SAGA CORE disruption.

Taken together, we propose a working model in which the structural integrity of the SAGA complex is dependent on its non-enzymatic CORE subunits, TAF5L, TAF6L and TADA1, with complex integrity essential for engagement and retention of the HAT module with SAGA, and consequently for the stability and function of its histone acetyltransferase, KAT2A (Fig. 6f). Disruption of the SAGA CORE leads to disengagement of the HAT from SAGA, potentially via disruption of co-translational or early post-translational assembly, and leads to the accumulation of non-complexed KAT2A protein that is not recruited to chromatin and is instead targeted for proteasomal degradation. This degradation is mediated, at least in part, by orphan quality control factors such as the E3 ligase UBR5 (Fig. 6f), in concert with the deubiquitinase OTUD5, which is known to regulate UBR5 stability. Interestingly, this degradation is mediated by paralogue-specific residues at the KAT2A N-terminus, which are absent in the closely related KAT2B protein. Thus, the stability and activity of KAT2A are tightly coupled to the structural organisation of the SAGA complex, and perturbing its assembly triggers a quality control response that selectively eliminates unassembled KAT2A, with functional consequences for histone acetylation and gene regulation. More broadly, our study highlights a general vulnerability in multi-subunit co-activator complexes that could be exploited therapeutically to target chromatin-modifying enzymes in cancer. By targeting structural complex components that themselves lack enzymatic activity, it is possible to disrupt complex function through two distinct yet complementary mechanisms. Loss of complex integrity impairs recruitment of enzymatic subunits to chromatin, leading to reduced catalytic output due to loss of interactions between the fully assembled complex and its substrate, while also leading to increased incidence of orphan complex subunits that are collaterally degraded in the absence of the intact complex, further reducing complex functionality.

## Discussion
Our study identifies a mechanism by which the structural integrity of the SAGA co-activator complex regulates the stability and function of one of its catalytic subunits, KAT2A. We demonstrate that a subset of non-enzymatic components of the SAGA CORE module, namely TADA1, TAF5L and TAF6L, are required to maintain KAT2A protein abundance and activity (Figs. 1 and 2). Loss of these structural subunits leads to disengagement of the HAT module, mislocalisation of KAT2A from chromatin (Figs. 2 and 3), and its subsequent degradation via the ubiquitin-proteasome system (Fig. 5) with concurrent, collateral loss of other elements of the CORE and HAT modules (Fig. 4). Our results also highlight a paralogue-specific degradation pathway (Fig. 6) centred on

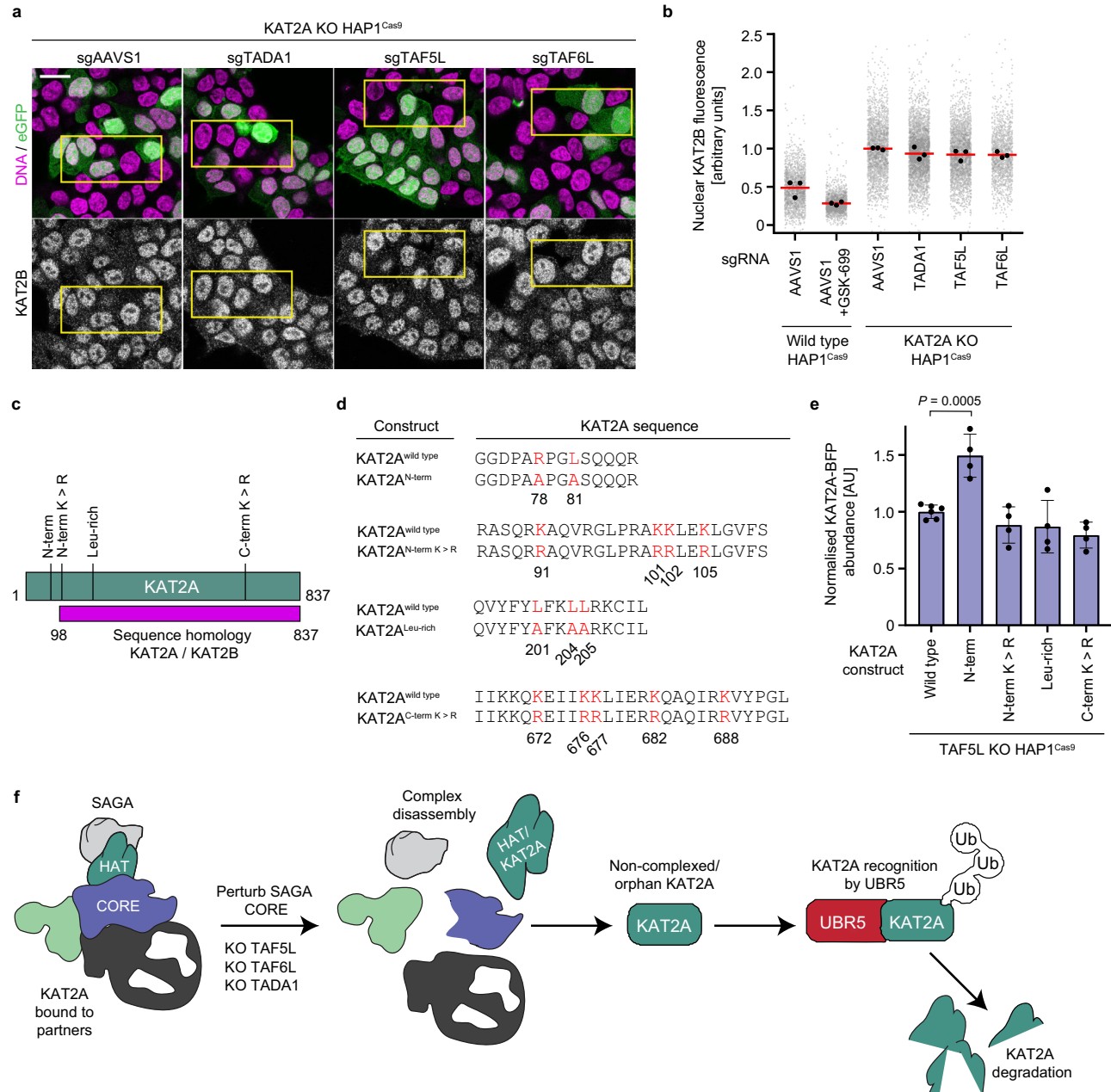

**Fig. 6 | Paralogue-specific residues destabilise KAT2A. a** Representative immunofluorescence images of KAT2B or DNA for KAT2A KO HAP1[Cas9] cells transduced with the indicated eGFP-expressing sgRNAs (cells fixed 5 days post-transduction). **b** Quantification of nuclear KAT2B fluorescence by immunofluorescence for KAT2A KO[Cas9] cells transduced with the indicated sgRNAs. **c** Schematic of KAT2A structure (green) and percentage of sequence homology of KAT2A with KAT2B (magenta). Putative residues that lead to KAT2A destabilisation are annotated. Numbers correspond to amino acid position. **d** Overview of KAT2A stability reporter constructs used in (**e**). Mutated residues and the corresponding wild-type residues are visualised in red. **e** Quantification of KAT2A-BFP abundance by flow cytometry for wild-type and mutant KAT2A reporter constructs in TAF5L KO HAP1[Cas9] cells. **f** Model of KAT2A destabilisation following SAGA CORE disruption. Loss of the SAGA CORE components TAF5L, TAF6L or TADA1 results in disassembly of the SAGA complex.

Complex disassembly leads to increased abundance of non-complexed KAT2A, converting it into a substrate that is recognised by the E3 ligase UBR5, funnelling KAT2A into the orphan protein degradation pathway. Biological replicates: **a**, **b** ($n = 3$); **e** ($n = 4$). Technical replicates: **e** ($n = 3$). **b** Grey dots represent the mean of individual nuclei. **b**, **e** Black dots represent the mean of each biological replicate. **b** Red bars indicate the mean of biological replicates. **e** Error bars represent standard deviation; bars indicate the mean for each condition. Significance was tested using a one-way ANOVA with a post-hoc Dunnett's multiple comparison test. Exact $p$ values are specified in the figure. All images show single Z-slices. Yellow boxes indicate inset regions showing non-transduced (eGFP-negative) and transduced (eGFP-positive) cells. To aid visualisation, the DNA and eGFP channels are not contrast-matched. Scale bar: 20 µm.

residues uniquely found at the KAT2A N-terminus, and absent in its paralogue KAT2B. Together, these findings are consistent with a mechanism in which, upon disassembly of SAGA, KAT2A exposes its N-terminus to the orphan protein quality control system, leading to its degradation. More broadly, our study reveals that SAGA-dependent

gene regulation relies not only on the enzymatic activities of its HAT and DUB modules but also on a finely tuned assembly architecture that protects vulnerable catalytic components from orphan degradation.

Our work builds on the emerging paradigm that many multi-subunit protein complexes are assembled co-translationally and

depend on precise stoichiometric balance to prevent the accumulation of orphan proteins[3,4]. By perturbing specific SAGA CORE components, we effectively uncouple KAT2A from its native assembly pathway, rendering it susceptible to proteasomal clearance. The gradual depletion of KAT2A following acute degradation of TADA1 suggests that CORE components may serve a chaperone-like function during or shortly after KAT2A translation, ensuring its successful incorporation into the SAGA complex. This process extends to other subunits that are directly interacting with TADA1, such as its co-translational assembly partner TAF12[7,36], or other subunits in close proximity to TADA1 in the CORE and HAT modules, including TAF5L, TAF6L and TADA3[21]. Altogether, these results show progressive disruption of SAGA complex integrity and HAT module engagement upon CORE perturbation, ultimately leading to an increased abundance of non-complexed KAT2A that is subsequently targeted for collateral protein degradation.

Importantly, we identify UBR5 and OTUD5, previously linked to orphan protein quality control and homeostatic turnover of key transcription factors such as MYC[13,59], as critical effectors of collateral KAT2A degradation. Although we observed a mild rescue in KAT2A protein levels upon proteasome inhibition or genetic disruption of *UBR5* or *OTUD5* in wild-type cells, the extent of rescue was substantially increased in cells lacking an intact SAGA CORE, demonstrating the importance of this degradation pathway when the SAGA CORE is disrupted. UBR5 is known to recognise a number of different degron motifs, often enriched in basic or hydrophobic residues, suggesting that once SAGA disassembles following CORE disruption, exposed degrons or misfolded regions within KAT2A are recognised by the orphan surveillance machinery, tagging the protein for proteasomal degradation. Although we provide evidence that KAT2A is able to associate to some extent with the ATAC complex when SAGA is unavailable following CORE disruption (Supplementary Fig. 7), our results indicate that this is insufficient to protect KAT2A from orphan quality control and collateral degradation mediated by UBR5.

Our functional data also highlights residues at the KAT2A N-terminus, absent in KAT2B, as a degron that leads to its destabilisation. A high-resolution structure of the human SAGA HAT, however, remains unavailable[21], while a recently published high-resolution HAT structure from *C.thermopilum*[62], including KAT2A, lacks the first 410 amino acids of the human KAT2A sequence, meaning the precise location of the identified degron and its potential interaction partners are not currently known. Further structural and biochemical studies will therefore be necessary to establish the nature of KAT2A's binding interactions, depending on whether it is SAGA-bound or not, and how this relates to its overall stability.

This work has broader implications for therapeutic strategies targeting multi-protein complexes in cancer. Indeed, our results indicate that KAT2A loss upon targeting the SAGA CORE applies more generally, as perturbation of CORE components resulted in substantial depletion of KAT2A protein not only in HAP1 cells, but also in cells derived from two different subtypes of leukaemia. While much attention has focused on inhibiting enzymatic activities within multi-subunit protein complexes such as SAGA, our data provides further evidence that destabilising critical structural subunits may offer an alternative approach, as targeting non-enzymatic scaffolding proteins could lead to secondary degradation of catalytic components, amplifying the therapeutic impact. Such a strategy could be particularly attractive in cancers dependent on SAGA-mediated acetylation programmes, as KAT2A has been identified as a cancer vulnerability in multiple contexts[30,34]. Indeed, collateral degradation of SAGA components could contribute to the dependencies observed in neuroblastoma and leukaemia[34,39].

In this context, small molecules targeting TAF5L, TAF6L or TADA1 would abrogate KAT2A function on two fronts: first, via proteasome-mediated collateral degradation, reducing KAT2A

protein abundance, and secondly, by disrupting complex integrity and chromatin binding. The latter is likely to affect not only KAT2A, but also its paralogue KAT2B, which, although not the predominant acetyltransferase found within SAGA, can be incorporated into SAGA instead of KAT2A[23,24]. Notably, our results indicate that targeting structural components of SAGA circumvents compensatory mechanisms involving the upregulation of KAT2B. Given the functional redundancy between KAT2A and KAT2B[34], targeting shared structural scaffold subunits therefore has the potential to simultaneously target both paralogues at the same time, an approach that could be exploited therapeutically in KAT2A-dependent cancers[30,34], in addition to or in combination with KAT2A/B degraders that have been developed[34,35,44], but not yet tested in patients.

Future work will be necessary to dissect how broadly this mechanism applies to other multi-subunit transcriptional regulators and to fully define the interactions between KAT2A and UBR5 that trigger its orphan degradation. It will also be important to determine whether additional components of the orphan quality control machinery are selectively engaged depending on the nature of the disrupted complex, and whether those can be specifically hijacked using proximity-inducing pharmacology. Given the essential role of transcriptional co-activators like SAGA in cell fate regulation, these insights could pave the way for novel approaches to target transcriptional addiction in cancer and other diseases.

## Methods

### Resource availability
All reagents generated in this study are available upon request to the lead contact with a completed materials transfer agreement. Requests for further information or reagents should be directed to the lead contact, D.S.

### Cell lines
All HAP1 cell lines used in this study are derived from a parental HAP1 cell line (Horizon Discovery C631). HAP1 cells were cultured in IMDM (Gibco). HEK293T cells were cultured in DMEM. NALM6, MV-411 and MOLM-13 cells were cultured in RPMI. All media were supplemented with 10% (v/v) foetal bovine serum (Gibco, 10270-106), 1% (v/v) Penicillin–Streptomycin (Fisher Scientific, 11548876) and L-Glutamine (Thermo Fisher, 25030024). Cells expressing Blasticidin or Puromycin resistance cassettes were cultured in medium supplemented with either 10 μg ml⁻¹ Blasticidin (Gibco, A1113903) or 1 μg ml⁻¹ Puromycin (Thermo Fisher Scientific, A1113803). All cell lines were cultured in a humidified incubator at 37 °C with 5% $CO_2$ and regularly tested negative for mycoplasma contamination. A list of cell lines used in this study can be found in Supplementary Data 2.

### Lentivirus production
HEK293T cells were seeded into 15 cm dishes ~24 h prior to transfection. Cells were transfected at 80% confluence in 16 ml of medium with 8.75 μg of VSV-G, 16.25 μg of psPAX2 and 25 μg of the CRISPR-Cas9 pooled lentiviral plasmids, using 150 μg of linear polyethylenimine (PEI) (Sigma-Aldrich). Medium was removed and replaced with fresh medium 16–24 h post-transfection. Lentiviral supernatant was collected 48 and 72 h post-transfection and subsequently concentrated by ultracentrifugation (98,000 × *g*, 4 °C, 2 h). For crude lentiviral supernatants, transfections were performed in a 6-well format. Briefly, cells were transfected at 80% confluence in 2 ml of media with 350 ng of VSV-G, 650 ng of psPAX2 and 1 μg of the lentiviral construct using 6 μg of PEI. Media was exchanged 16–24 h post-transfection. The supernatant was collected 48 and 72 h post-transfection as separate harvests and centrifuged to remove cellular debris (4000 × *g*, 4 °C, 10 min). For HAP1 cells, crude lentiviral supernatant was diluted with complete medium before transducing cells, as specified below (section: Arrayed CRISPR screening using the KAT2A stability reporter).

For transductions of NALM6 cells, 100 µl of crude lentiviral supernatant was diluted 1:1 with complete medium and added to the cells. For transductions of MV4-11 and MOLM-13 cells, RPMI was supplemented with Polybrene (4 µg/ml) and spinfection was performed at 1000 × *g* for 90 min at 37 °C.

## Generation of Cas9-expressing cells

To generate HAP1 cells stably expressing Cas9 in both wild type and TAF5L KO backgrounds, in addition to the leukaemia lines MOLM-13, NALM6 and MV4-11, cells were transduced with Cas9-Blast lentivirus (generated from pLentiCas9-Blast, Addgene, 52962) and selected with 10 µg ml$^{-1}$ Blasticidin (Gibco, A1113903) for 7 days. Clones were obtained by limited dilution into 96-well plates. Cas9 activity was determined with a competition assay using sgRNAs targeting either the essential genes *RAD21* or *SF3B5*, or the safe harbour locus *AAVS1*, cloned into an eGFP-expressing vector (pLKO5.gRNA.EFS.GFP (Addgene, 57822)). eGFP fluorescence was measured 3 and 7 days (HAP1) or 3, 6, 9 and 13 days (leukaemia lines) post-transduction by flow cytometry using an LSR Fortessa instrument (BD Biosciences) with an HTS Loader. Clones showing dropout of eGFP fluorescence upon targeting *RAD21* or *SF3B5* but not *AAVS1* were selected.

## Generation of HAP1 knockout clones

sgRNAs were designed to target critical exons of the selected candidate genes and cloned into pLKO5.gRNA.EFS.GFP. One day before transfection, 400,000 HAP1$^{Cas9}$ cells were seeded into 6-well plates. The following day, cells were transfected with 2 µg sgRNA plasmid in total (1 µg per sgRNA plasmid if using multiple plasmids) using Turbofect Transfection Reagent (Thermo Fisher Scientific, R0531). Two to three days post-transfection, eGFP-positive cells were sorted using a FACSAria Fusion instrument (BD Biosciences) and replated into 6-well plates. Cells were allowed to recover in fresh complete medium for 1 day. The following day, cells were detached from the culture plate and seeded at clonal density into 96-well plates. Following clonal expansion, individual clones were genotyped by PCR and Sanger sequencing to identify homozygous/biallelic deletion or knockout (KO) clones. Primers used for genotyping are listed in Supplementary Data 3.

## Generation of KAT2A stability reporter expressing cells

The KAT2A stability reporter was generated by modifying a previously published BRD4 reporter plasmid (Hsia et al.[43]). The BRD4 cDNA sequence was excised and replaced with that of KAT2A by SalI/BamHI digestion and Gibson Assembly. The KAT2A ORF was PCR-amplified from TFORF2959 (Addgene, 144435). The resultant pRRL-SFFV-KAT2A-GGGG-3xV5-TagBFP-P2A-mCherry plasmid was produced as crude lentivirus as described above and used to transduce HAP1 cells. Seven to ten days post-transduction, cells were sorted for mCherry-positive cells using a FACSAria Fusion instrument (BD Biosciences). Cells expressing low, medium and high levels of mCherry were sorted. For the arrayed loss-of-function CRISPR screen performed in wild-type HAP1$^{Cas9}$ cells, cells expressing high levels of mCherry were used. For the pooled gain-of-function CRISPR screen performed in TAF5L KO$^{Cas9}$ cells and validation experiments using inhibitors, cells expressing low to medium levels of mCherry were used. KAT2A reporter constructs with mutations at putative degron motifs were obtained by gene synthesis (Twist Bioscience).

## Arrayed CRISPR screening using the KAT2A stability reporter

Crude lentiviral supernatants for sgRNAs were generated as described above and distributed in an arrayed format in 96-well plates. One day prior to transduction, wild-type HAP1$^{Cas9}$ cells expressing the KAT2A stability reporter, in addition to the parental cell line without the reporter, were seeded into F-bottomed 96-well plates (VWR, 734-2327) at 7000 cells per well in complete IMDM. For seeding, cells were

washed once with PBS and incubated with Accutase® Cell Detachment Solution (Thermo Fisher, 423201) for 3–5 min at room temperature, quenched with complete medium, and filtered once through a 35 µm nylon mesh before counting. On the day of transduction, cells were transduced with 100 µl of crude viral supernatant (50 µl of crude supernatant diluted 1:1 with complete medium), with one sgRNA transduced per well (in biological triplicates). Both the parental cell line and reporter expressing cells were transduced with each sgRNA in the library. Three, five and seven days after transduction, half of the cells in each well were analysed by flow cytometry. The remaining half were passaged, with fresh complete medium added to a final volume of 200 µl per well. For flow cytometry, the medium was removed from each well, cells were washed twice with 100 µl PBS, and incubated with 100 µl Accutase for 5–10 min at room temperature. Cells were resuspended in the plate and 50 µl of the cell suspension transferred to a V-bottomed 96-well plate (Thermo Fisher, 277143) containing 50 µl of FACS buffer (PBS containing 2% FCS, 2 mM EDTA, pH 8.0 (Thermo Fisher, AM9260G)). BFP, eGFP and mCherry fluorescence was measured for each well using an LSR Fortessa instrument (BD Biosciences) with attached HTS Loader. Flow cytometry data were analysed using FlowJo V10.10.0. To calculate the sgRNA dropout for each gene, the fraction of eGFP-positive cells was determined for each guide at each time point. The data were normalised independently for each guide relative to the fraction of eGFP-positive cells 3 days after transduction. To calculate KAT2A-BFP protein abundance 5 days after lentiviral transduction, cells were first gated for eGFP-positive cells (to select only transduced cells) and then mCherry-positive cells (to select only cells expressing the reporter). To account for different transduction efficiencies between different eGFP-expressing sgRNAs, BFP and mCherry mean intensity values were normalised by background subtraction of the respective values for reporter-negative cells transduced with the same sgRNA. Relative KAT2A abundance was calculated as the ratio of background-subtracted KAT2A-BFP to mCherry mean fluorescence intensity for each sgRNA. Wild-type HAP1$^{Cas9}$ cells expressing the KAT2A stability reporter were gated in the following way: (1) FSC-A vs SSC-A to gate cells, (2) FSC-A vs FSC-H and SSC-H vs SSC-W to gate single cells, and FSC-A vs GFP-A to gate eGFP-positive cells. The KAT2A-BFP/mCherry ratio was measured for eGFP-positive cells for each guide. An example of the gating is shown in Supplementary Fig. 1a. This gating was used in Fig. 1c, Supplementary Figs. 2a–c, 3a, d–f, h, 4b–e and 7c. For stability reporter experiments in Figs. 5c and 6e, Supplementary Fig. 9d, the same gating strategy was used, with the exception of gating for eGFP-positive cells, as the experiment was performed on eGFP-negative cells. In Supplementary Fig. 9j, cells were gated for iRFP-positive rather than eGFP-positive cells. A list of the plasmids used and the sequences of the sgRNAs can be found in Supplementary Data 1.

## Compound treatment in live cells

Unless otherwise stated, compounds used in this study were prepared in complete medium for the corresponding cell line and used at the following concentrations: GSK-699 (Invivochem, V41244) was used at 100 nM for 6 h. MG-132 (MedChemExpress, HY-13259) and MLN-4924 (MedChemExpress, HY-70062) were used at 1 µM for 6 h. TAK-243 (MedChemExpress, HY-100487) was used at 500 nM for 6 h. dTAG$^{v}$-1 (Tocris, 6914) was used at 100 nM, with the duration of incubation specified in the corresponding figure. The equivalent concentration of DMSO (Fisher Scientific, BP231-100) was used in control cells.

## Pooled CRISPR screen using the KAT2A stability reporter

A FACS-based CRISPR/Cas9 screen using KAT2A stability reporter cells was performed, followed by library preparation, NGS, and data analysis, as previously described with minor modifications[38,43,52]. Briefly, a Ubiquitin-proteasome system-focused sgRNA library targeting 1301 human ubiquitin-related genes with 6 sgRNAs per gene[52]

was used to transduce TAF5L KO HAP1[Cas9] KAT2A-BFP-P2A-mCherry cells. Transductions were performed at a multiplicity of infection (MOI) of 0.1, ensuring a 1,000-fold library coverage. Ten days after G418 selection (1 mg ml$^{-1}$; Sigma-Aldrich, A1720), 50 million cells per replicate were harvested (three replicates harvested). Following selection, cells were incubated for 10 min at 4 °C with an anti-Thy1.1–APC antibody (BioLegend, 202526; 1:400) and Human TruStain FcX™ Fc Receptor Blocking Solution (BioLegend, 422301; 1:400). Cells were fixed with 4% BD Fixation Buffer (Thermo Scientific™ Pierce™, BD 554655) for 45 min at 4 °C, protected from light, and stored overnight at 4 °C in PBS containing 5% FBS and 1 mM EDTA. The next day, cells were sorted on a BD FACSAria™ Fusion (BD Biosciences) using a 70 μm nozzle. Aggregates, dead cells and sgRNA library–negative (Thy1.1–APC–negative) cells were excluded. The remaining population was sorted into KAT2A-BFP$^{high}$ (~3% of cells), KAT2A-BFP$^{mid}$ (~30%) and KAT2A-BFP$^{low}$ (~3%) fractions based on KAT2A-BFP and mCherry expression, ensuring a minimum of 1500-fold library coverage per replicate. Genomic DNA from sorted cell populations was extracted by cell lysis (10 mM Tris-HCl, 150 mM NaCl, 10 mM EDTA, 0.1% SDS), followed by proteinase K digestion (New England Biolabs, P8107) and RNA removal using DNase-free RNase A (Thermo Fisher Scientific, EN0531). DNA was purified by two rounds of phenol extraction (Sigma-Aldrich, P4557) and isopropanol precipitation (Sigma-Aldrich, I9516). The sgRNA cassettes were then amplified in two PCR steps using AmpliTaq Gold polymerase (Thermo Fisher Scientific, 4311818), with sample-specific barcodes incorporated in the first PCR and Illumina adaptors added in the second. PCR products were cleaned using Mag-Bind TotalPure NGS beads (Omega Bio-tek, M1378-00), pooled and sequenced on an Illumina NovaSeq 6000 platform. The screen data were processed using the crispr-process-nf Nextflow pipeline [https://github.com/ZuberLab/crispr-process-nf/]. In brief, raw FASTQ files were trimmed with cutadapt (v4.4) to remove random barcodes and spacer sequences, followed by demultiplexing based on sample barcodes. Reads were aligned to the custom UPS sgRNA reference library using Bowtie2 (v2.4.5), and sgRNA abundance was quantified with featureCounts (v2.0.1). The resultant count tables (Supplementary Data 4) were analysed using the crispr-mageck-nf workflow [https://github.com/ZuberLab/crispr-mageck-nf/] for downstream statistical evaluation. Gene-level enrichment was assessed using MAGeCK (v0.5.9), comparing KAT2A-BFP$^{high}$ or KAT2A-BFP$^{low}$ populations to the KAT2A-BFP$^{mid}$ reference group, based on median-normalised read counts and replicate-level variance estimation.

## Immunofluorescence

For cell seeding, cells were washed once with PBS and incubated with Accutase for 3–5 min at room temperature, quenched with complete medium and filtered once through a 35 μm nylon mesh before cell counting. For confocal microscopy experiments, cells were seeded at 50,000 cells per well 1 day prior to fixation in 8-well Ibidi chambers (Ibidi, 80827). For arrayed CRISPR screening experiments in 96-well plates, 7000–10,000 cells were seeded per well 1 day prior to transduction in F-bottomed 96-well plates (VWR, 734-2327). The following day, cells were transduced with 100–150 μl of crude viral supernatant (crude viral supernatant diluted 1:1 with complete medium). Three days after transduction lentivirus-containing medium was removed. Cells were detached from the plate using Accutase and transferred to imaging plates (PhenoPlate 96-well plates) (black, optically clear flat-bottom, tissue-culture treated (Revvity, 6055302)) or 8-well Ibidi chambers. Cells were fixed for immunofluorescence 5 days (KAT2A, KAT2B staining) or 6 days (H3K9ac staining) post-lentiviral transduction. At the time of fixation, cells were washed twice with PBS before fixation with PBS containing 4% methanol-free formaldehyde (Thermo Fisher Scientific, 28906) for 5 min (Ibidi chambers) or 15 min (96-well plates). The fixative was removed, and cells quenched using 10 mM

Tris-HCl (Invitrogen, 15567027), pH 7.5 in PBS for 3 min. Cells were permeabilised with PBS containing 0.2% Triton X-100 (Sigma-Aldrich, X100-100 ml) for 5 min or 0.5% Triton X-100 for 15 min (H3K9ac staining experiments only), washed once with PBS to remove residual detergent, and blocked for 30 min with 0.45 μm filtered 2% BSA (Sigma-Aldrich, A9418-50G) in PBS (blocking buffer). Primary and secondary antibodies were diluted with blocking buffer. Primary antibody incubations were performed for >16 h at 4 °C with gentle shaking, followed by 3 × 10 min washes with PBS, and incubation with secondary antibodies for 1 h at room temperature with gentle shaking. Samples were then washed 3 × 10 min with PBS containing 1.62 μM Hoechst 33342 (Thermo Fisher Scientific, H3570). Antibodies used for immunofluorescence and their working dilutions can be found in Supplementary Data 2.

## Microscopy and image processing

All confocal microscopy experiments, with the exception of Fig. 1d, e, were imaged at room temperature on a custom Zeiss LSM 980 microscope fitted with an additional Airyscan2 detector, using a 63 × NA 1.4 oil DIC Plan-Apochromat (Zeiss) objective and ZEN 3.3 Blue 2020 software. Images on the LSM 980 were collected at 1908 × 1908 pixels with a spatial sampling of 0.0706 μm per pixel. The data in Fig. 1d, e, were acquired on an Opera Phenix high-content confocal microscope (PerkinElmer) using a 20 × air objective and imaging 18 fields of view per well with incubation at 37 °C with 5% CO$_2$. Images on the Opera Phenix high-content system were collected at 1080 × 1080 pixels with a spatial sampling of 0.5926 μm per pixel. Images on both systems were acquired as single optical sections. For images acquired using the Opera Phenix system, individual channels were flat-field corrected using Python following acquisition. The flat-field corrected images were then merged into multichannel composite images, which were used as the input for the analysis pipeline. Images were prepared for visualisation using Fiji (NIH, version 2.14.0). For image visualisation, the representative images in the requisite figures are contrast-matched relative to the condition with the brightest signal for all channels, where quantitative comparisons are made (KAT2A, KAT2B, H3K9ac). Brightness and contrast were adjusted linearly across the entire image for the channel of interest. These settings were then applied uniformly across all images shown in the figure. The DNA channel was contrast-matched only for Fig. 2d, where quantitative assessment of DNA fluorescence was necessary. Otherwise, the DNA and eGFP channels were not contrast-matched to aid visualisation. This is noted in the requisite figure legends.

## Image analysis—measurement of mean pixel intensities in fields of cells

Fields of cells were analysed using a custom automated analysis pipeline written in Python. Before measuring the mean nuclear fluorescence of markers of interest, the following criterion was pre-established: only interphase cells were considered for the analysis. For most immunofluorescence experiments, H3S10P was also stained as a mitotic marker, with H3S10P-positive cells excluded from the analysis. Size and maximum mean DNA fluorescence filters were applied to the data to also help exclude mitotic cells. A Gaussian blur was applied to the DNA (Hoechst 33342) channel using the skimage filters module, and the DNA channel was then segmented using the 'nuclei' module of cellpose. Cells which touched the border of the image were excluded using the clear_border functionality from the skimage segmentation module. Individual masks were labelled and applied to each cell in the field. The area of the nuclear mask and mean fluorescence within the nuclear mask were then calculated for each segmented cell, and the data were output as a csv file for each input condition. Biological replicates were normalised independently, relative to their own internal positive and negative controls, and the normalised data were then merged for the final figure. To account for differences in cell

ploidy and the local compaction state of the chromatin, in Fig. 2d, the H3K9ac signal level for each condition was determined by dividing the mean nuclear H3K9ac fluorescence of each segmented nucleus by the mean fluorescence of the DNA (stained with Hoechst 33342), to calculate a H3K9ac/Hoechst ratio.

## RNA isolation and RT-qPCR
RNA was isolated from cells using a RNeasy Mini Kit (Qiagen, 74104) according to manufacturer instructions. 500 ng RNA was used for cDNA preparation with an iScript cDNA Synthesis Kit (Bio-Rad, 170889). RT-qPCR was performed using an iQ SYBR Green Supermix (Bio-Rad, 1708880) on a Bio-Rad iCycler RT-PCR detection system. In Supplementary Figs. 6a, d and 9b, c, KAT2A or KAT2B transcript levels were normalised to GAPDH. Primers used for RT-qPCR are listed in Supplementary Data 5.

## CUT&RUN sample preparation
For each sample, nuclei from 250,000 cells were isolated in 100 µl nuclear extraction (NE) buffer (20 mM HEPES) (Thermo Fisher, 15630-056), 10 mM KCl (Thermo Scientific, AM960-G), 0.1% Triton X-100 (Sigma, 93443-100 ML), 20% Glycerol (Fisher Scientific, 10692372), 0.5 mM Spermidine (Sigma, S2626-1G) supplemented with Roche cOmplete EDTA-free Protease Inhibitors (Sigma, 11873580001) for 10 min on ice, centrifuged for 3 min at $600 \times g$ at room temperature, resuspended in 100 µl NE buffer and bound to 10 µl of activated ConA beads for 10 min at room temperature. Next, beads were collected with a magnetic rack and resuspended in 50 µl Antibody Buffer (20 mM HEPES, 150 mM NaCl (Fisher Scientific, BP358-1), 0.5 mM Spermidine, 0.02% Digitonin (Millipore, D141-100MG), 2 mM EDTA supplemented with Roche Complete EDTA-free Protease Inhibitors) supplemented with 1 µl of antibody. Nuclei were incubated overnight on a nutator at 4 °C. After two washes with 250 µl Digitonin buffer (20 mM HEPES, 150 mM NaCl, 0.5 mM Spermidine, 0.02% Digitonin supplemented with Roche Complete EDTA-free Proteasome Inhibitor), nuclei were resuspended in 50 µl Digitonin buffer supplemented with 700 ng/µl pAG-Mnase (gift from Joris Van der Veeken, IMP, Vienna), and incubated for 30 min at room temperature. Excess pAG-Mnase was removed with two washes with Digitonin buffer. Finally, nuclei were resuspended in 50 µl Digitonin buffer, and pAG-Mnase was activated with 1 µl 100 mM CaCl₂ for 1.5 h on ice. The reaction was quenched with 33 µl Stop Buffer (340 mM NaCl, 20 mM EDTA, 4 mM EGTA (VWR, J60767AD)), 50 µg/mL RNAse A, 50 µg/mL Glycogen (Merck, 10901393001). Cleaved DNA was allowed to diffuse out by incubating the tubes at 37 °C for 10 min. After separation using a magnetic rack, DNA was isolated using the Monarch DNA Cleanup Kit (NEB, T1030) and eluted in 15 µl Elution Buffer. Next-generation sequencing libraries were prepared from 2 ng DNA using the NEBNext Ultra II DNA Library prep Kit (NEB, E7645L), pooled and sequenced (PE150) at Azenta, at a depth of 5-10 M reads per library. Antibodies used for CUT&RUN and their working dilutions can be found in Supplementary Data 2.

## CUT&RUN data analysis
Sequencing reads (150 bp paired-end, NovaSeq 6000) were trimmed using TrimGalore and aligned to the human genome, hg38, using Bowtie2[63] (--local, --very-sensitive, --no-mixed, --no-discordant, --dovetail). SAMtools[64] was used to fix mates of paired-end reads, merge, sort and index bam files, while a CUT&RUN-specific blacklist[65] was removed from the bam files using BEDtools[66]. DeepTools[67] was then used to build the genome coverage, to compute matrices and plot the profiles of KAT2A or H3K9ac aligned around TSSs.

## Sucrose gradient fractionation
Sucrose (Fisher Scientific, 15446759) stock solutions (10% and 60%, prepared in dH₂O), were diluted 1:1 with 2X NP40-Lysis buffer (40 mM Tris-HCl pH 7.5 (Invitrogen, 15567027), 200 mM NaCl (Fisher Scientific, BP358-1), 10 mM MgCl₂ (Merck, MG1028-100ML), 0.4% NP40 (Thermo

Fisher, 85124)) to generate 5 and 30% sucrose solutions in 1X NP40-Lysis buffer. DTT (Sigma, 43816-10ML) was added to a concentration of 1 mM to diluted sucrose solutions immediately before gradient preparation. 5–30% sucrose gradients were prepared in ultracentrifuge tubes (Open-Top Polyclear Centrifuge Tubes, 14 × 89 mm, Seton Scientific, 7030) using a BioComp gradient station (Programme: Long cap, Sucrose, 05–30%, weight/volume), and subsequently equilibrated to 4 °C for >1 h before sample loading. The rotor was also equilibrated to 4 °C before ultracentrifugation. Samples were harvested freshly before ultracentrifugation. Cells growing in 15 cm plates were harvested by washing once with ice-cold PBS, with banging against a solid support to remove dead cells. PBS was removed, and cells were detached by incubation with Accutase for 3–5 min at room temperature. Cells were then quenched with complete medium, centrifuged ($400 \times g$, 5 min, 4 °C), washed once with ice-cold PBS, centrifuged once more and transferred to 1.5 ml Eppendorf tubes in 1 ml ice-cold PBS. Cells were centrifuged ($1100 \times g$, 1 min, 4 °C), PBS removed and cell pellets lysed in 150 µl 1X NP40-lysis buffer (20 mM Tris-HCl pH 7.5, 100 mM NaCl, 5 mM MgCl₂, 0.2% NP40, 1 mM DTT, 2X Roche cOmplete EDTA-free protease inhibitors (Sigma, 11873580001)) for 30 min on ice, with vortexing every 10 min. Lysates were clarified by centrifugation ($17,000 \times g$, 15 min, 4 °C), and the concentration of the clarified supernatant was quantified using a BCA kit (Pierce, Thermo Scientific, 23227). Sample concentrations and volumes were normalised to load 300 µg protein onto each sucrose gradient, with 30 µg of each sample set aside as inputs. Ultracentrifugation was performed for 16 h at $158,000 \times g$ at 4 °C using an SW-41-Ti rotor on a Sorvall WX+ Ultra Series instrument. Following ultracentrifugation, samples were fractionated into 1.5 mL Eppendorf tubes using a BioComp Gradient Fractionator. Fractions were subsequently concentrated using Trichloroacetic (TCA) precipitation. 100% TCA (Sigma, 92128-100 G) was added to each fraction (10% of the fraction volume added), the samples were gently inverted to mix, and the samples were incubated on ice for 15 min. Samples were then centrifuged at high speed ($17,000 \times g$, 10 min, 4 °C), the supernatant removed and samples incubated with 500 µL ice-cold 100% acetone (Roth, 9372.1) for 10 min on ice. Samples were once more centrifuged at high speed, the supernatant removed and washed once more with ice-cold acetone. After centrifugation and removal of the supernatant, the resultant pellet was resuspended in 2X Laemmli buffer (containing 20% Tris-HCl, pH 8.0 (Thermo Fisher, AM9855G)) and boiled for 10 min at 95 °C, before analysis of the fractions by western blotting.

## Western blotting
Protein extracts were mixed with Laemmli buffer, boiled for 10 min at 95 °C, and centrifuged for 1 min at $13,000 \times g$ to remove aggregates. Samples were resolved on a 4–12% gradient Bis-Tris gel (Bio-Rad, 3450125) and transferred to a PVDF membrane using the Trans Blot Turbo system (Bio-Rad). Blots were incubated in blocking buffer (5% milk in PBS-T) (0.05% Tween 20 (Sigma, P9416-100ML)) for >30 min at room temperature with shaking. Primary antibodies were diluted in blocking buffer and incubated for >16 h at 4 °C with shaking. Following primary antibody incubation, samples were then washed 3 × 15 min with PBS-T at room temperature with shaking, before incubation with an anti-rabbit HRP-conjugated secondary antibody (Cell Signaling, 7074S) at 1:10000 dilution for 1 h at room temperature. Samples were then washed 3 ×15 min with PBS-T. Blots were developed by incubation with Clarity Max™ ECL Substrate (Bio-Rad, 1705062) and then visualised on a Bio-Rad Chemidoc imager. Antibodies used for western blotting and their working dilutions can be found in Supplementary Data 2.

## TMT-expression proteomics sample preparation–FASP + C18 cleanup
Cells were lysed in lysis buffer (2% SDS, 50 mM HEPES, 1 mM PMSF, 1 x protease inhibitor cocktail from Roche). Samples were resuspended

and incubated at room temperature for 20 min before they were sonicated in a Branson ultrasonic processor with a microtip on ice (0.5 s on / 0.5 s off, 30 s total, 20% input). The lysate was centrifuged at 16,000 × g for 10 min at 20 °C, and the supernatants were transferred to a fresh tube. The protein concentration was measured with a Pierce BCA Protein Assay (Product Nr.: 23227) according to the manufacturer's instructions. The FASP protocol was employed for proteolytic digestion as previously reported[68] with slight modifications. Briefly, DTT (83.3 mM) was added to the lysis buffer, and the sample was incubated at 95 °C for 5 min. Microcon 30 Ultracel YM-30 filter units were primed once with 200 μL urea buffer (8 M urea in 100 mM Tris/HCl, pH 8.5). Samples (100 μg) were then loaded onto the filter units at 14,000 × g at 20 °C for 15 min, washed with UA solution (8 M Urea in 100 mM Tris/HCl, pH 8.5). Alkylation was performed with 200 μl 50 mM Iodacetamide in UA solution. Samples were again centrifuged (14,000 × g, 10 min) after a 30-min incubation period in the dark. Next, samples were washed twice with 100 μl UA and then twice with 100 μl 50 mM TEAB. Samples were then digested in 40 μl 50 mM TEAB, pH 8.5, at a protein to enzyme ratio of 50:1. The spin columns were sealed and incubated at 37 °C overnight. The next day, samples were spun down (14,000 × g, 20 min) and the columns were washed once with 40 μL 50 mM TEAB and once with 50 μL 0.5 M NaCl. The combined filtrate was acidified with 30% TFA until a pH below 3 was achieved. Samples were then subjected to C18 cleanup with Pierce™ Peptide Desalting spin columns according to the manufacturer's instructions (Thermo Fisher Scientific; Catalogue numbers 89851, 89852).

## TMT-expression proteomics—peptide quantification

Desalted peptides were reconstituted in LC-grade water, and peptide concentrations were determined using the Pierce™ Quantitative Fluorometric Peptide Assay (Thermo Fisher Scientific, Ref: 23290) according to the manufacturer's instructions. Fluorescence measurements were performed on a SpectraMax® i3x Multi-Mode Microplate Reader (Molecular Devices).

## TMT-expression proteomics—TMT labelling

Peptides were subsequently buffered to a final concentration of 50 mM HEPES, pH 8.5. Labelling was performed at a peptide concentration of ≥1 μg/μL using TMT reagent prepared in anhydrous acetonitrile at 16.7 μg/μL, applied at a TMT-to-peptide mass ratio of approximately 2:1 to 3:1. The final acetonitrile concentration during the labelling reaction was 35% (v/v). Samples were incubated for 1 h at room temperature with gentle agitation. Labelling reactions were subsequently quenched by the addition of 5% hydroxylamine solution in HEPES buffer to achieve a final hydroxylamine concentration of 0.2–0.4%, followed by incubation for 15 min at room temperature. Labelled peptides were pooled, dried by vacuum centrifugation, and reconstituted in 0.1% trifluoroacetic acid for subsequent processing. Labelling efficiency was routinely assessed following final analysis.

## TMT-expression proteomics—offline fractionation of TMT pool

Pooled TMT labelled sample was concentrated and desalted by using a Pierce Peptide Desalting Spin Columns (Catalogue Number 89851 and 89852) according to manufacturer's instructions with minor modifications: Peptide elution was done in 2 steps: 40% ACN, 100 mM TEAB and 70% ACN, 100 mM TEAB solution. Eluates were pooled and brought to dryness in a SpeedVac vacuum concentrator. Dry peptides were resuspended in 30 μl 10 mM ammonium formate (pH 10) and fractionated by reverse-phase chromatography at basic pH by using a Gemini-NX C18 (150 × 2 mm, 3 μm, 110 A) column (Phenomenex, Torrance, USA) on an Ultimate 3000 RSLC micro system (Thermo Fisher Scientific) equipped with a fraction collector. Peptides were separated at a flow rate of 50 μl/min in 10 mM ammonia formate buffer (pH 10.0) and eluted over a 70 min nonlinear gradient from 0 to 100% solvent B

(90% acetonitrile, 10 mM ammonium formate, pH 10.0). Thirty-six concatenated fractions were collected in a time-based manner (at 30 s intervals) between minutes 11.5 and 57. Fractions were immediately acidified by adding 5 μl 30% TFA. After fractionation, organics were removed in a vacuum concentrator, and samples were stored at −20 °C until MS analysis.

## TMT-expression proteomics—LC-ESI-MS/MS data acquisition

For MS analysis, fractions were reconstituted in 150 μl 0.1% TFA and 5 μl was injected onto the machine. For proteomic data acquisition, a nanoflow LC–ESI-MS/MS setup comprised of a Dionex Ultimate 3000 RSLCnano system coupled to a Fusion Lumos mass spectrometer (both Thermo Fisher Scientific Inc.) was used in positive ionisation mode. MS data acquisition was performed in data-dependent acquisition (DDA) mode. For proteome analyses, peptides were delivered to a trap column (Acclaim™ PepMap™ 100 C18, 3 μm, 5 × 0.3 mm, Thermo Fisher Scientific) at a flow rate of 5 μL/min in HPLC-grade water with 0.1% (v/v) TFA. After 10 min of loading, peptides were transferred to an analytical column (ReproSil Pur C18-AQ, 3 μm, Dr. Maisch, 500 mm × 75 μm, self-packed) and separated using a stepped gradient from minute 11 at 8% solvent B (0.4% (v/v) FA in 90% ACN) to minute 61 at 28% solvent B and minute 81 at 40% solvent B at 300 nL/min flow rate. The nano-LC solvent A was 0.4% (v/v) FA HPLC-grade water. MS1 spectra were recorded at a resolution of 60,000 using an automatic gain control target value of $4 \times 10^5$ and a maximum injection time of 50 ms. The cycle time was set to 2 s. Only precursors with charge state 2–6, which fall in a mass range between 360 and 1300 Da were selected, and dynamic exclusion of 30 s was enabled. Peptide fragmentation was performed using higher energy collision dissociation (HCD) and a normalised collision energy of 35%. The precursor isolation window width was set to 1.3 m/z. MS2 spectra were acquired at a resolution of 30,000 with an automatic gain control target value of $5 \times 10^4$ and a maximum injection time of 54 ms. Using synchronous precursor selection, the top 10 fragment ions of the MS2 scans were isolated and subjected to HCD fragmentation in the linear ion trap using 55% normalised collision energy. MS3 spectra were acquired in the Orbitrap at a resolution of 50,000 over a scan range of 100–1000 Th, using an AGC target of 1e5 and a maximum injection time of 120 ms. The reagent tag type in the filer IsobaricTagExclusion was set to TMTpro.

## TMT-expression proteomics—data analysis

For all DDA measurements, MaxQuant (version 2.6.7.0) with its built-in search engine Andromeda[69,70] was used for peptide identification and quantification. MS2 spectra were searched against all Swiss-Prot canonical protein sequences obtained from UniProt (UP000005640, downloaded: 19 March 2025), supplemented with common contaminants (built-in option in MaxQuant). Trypsin/P was specified as the proteolytic enzyme. Precursor tolerance was set to 4.5 ppm, and fragment ion tolerance to 20 ppm. The minimal peptide length was defined as seven amino acids, and the 'match-between-run' function was disabled. Quantification was performed on the MS3 level. '18plex' (TMTpro) was selected as the isobaric labels, and the reporter mass tolerance was set to 0.003 Da. For proteome analyses, carbamidomethylated cysteine was set as a fixed modification and oxidation of methionine and N-terminal protein acetylation as variable modifications. The FDR was set to 100%. These search results were then used as input files for Oktoberfest (v.0.8.3)[71]. We performed Prosit rescoring[72] and quantification via the picked-group-FDR approach (v.0.8.1)[73]. The Prosit models Prosit_2020_irt_TMT for retention time prediction and Prosit_2020_intensity_TMT for intensity prediction were employed. The output files were filtered at 1% FDR. Perseus was used for data analysis[74]. Briefly, 'common contaminants', 'reversed' and 'only identified by site' were filtered out, the intensities for each TMT channel were log2-transformed, and median-centric normalised. Samples were

then categorically annotated to group replicates together, and the 'Hawaii plot' function with default values was performed. The obtained matrix was exported for further analyses.

## Statistics and data reporting

Statistical analyses were performed in Python using the modules scipy, statsmodels and scikit-posthocs. All statistical tests were performed on biological replicates. A one-way ANOVA was used to compare multiple groups with a significance threshold of 0.05. When the ANOVA was significant, a Dunnett's multiple comparison test was performed to compare conditions with a reference condition. The number of replicates and statistical test used for each experiment is specified in each figure legend. Exact $p$ values are specified in the figures. When performing pairwise comparisons, an unpaired, two-tailed Welch's t-test was used.

In Fig. 1, a one-way ANOVA was used to compare the effect of SAGA gene knockout on KAT2A-BFP abundance (c) or endogenous KAT2A fluorescence (e). After confirming significance, a Dunnett's multiple comparison test was performed to compare each condition with the sgNEG reference. In Fig. 1c, e, the mean of the negative control guides was calculated for each replicate to generate one negative control value per replicate. In Fig. 1c, sgKAT2A was excluded from the statistical analysis due to the reduction of both BFP and mCherry fluorescence when using anti-KAT2A guides against the reporter, as well as the splicing components SF3B3 and SF3B5, due to substantial eGFP dropout already 5 days after transduction. In Fig. 1c, e, only eGFP-positive cells were analysed, and data were normalised relative to sgAAVS1 transduced cells treated with the KAT2A/KAT2AB PROTAC GSK-699 (100 nM) for 6 h.

In Fig. 2, a one-way ANOVA was used to compare the effect of cDNA overexpression in the respective knockout cell line on nuclear KAT2A fluorescence (b), sgRNA knockout on H3K9ac levels (d), or the levels of nuclear KAT2B fluorescence in different HAP1 cell lines (j). After confirming significance, a Dunnett's multiple comparison test was performed to compare each condition with the wild type (b, j) or sgAAVS1 (d) reference. In Fig. 2b, j the data were normalised relative to the median nuclear KAT2A fluorescence of wild-type HAP1 cells. In Fig. 2d, only eGFP-positive cells were analysed.

In Fig. 4, a one-way ANOVA was used to compare the effect of acute degradation of TADA1-dTAG-HA on nuclear TADA1-HA fluorescence (b) or nuclear KAT2A fluorescence (d). After confirming significance, a Dunnett's multiple comparison test was performed to compare each condition with the DMSO-treated reference. In Fig. 4b, d, the data were normalised relative to the median nuclear TADA1-HA (b) or KAT2A (d) fluorescence of DMSO-treated cells.

In Fig. 5, a one-way ANOVA was used to compare the effect of inhibitors on nuclear KAT2A fluorescence (b) or KAT2A-BFP abundance (c) as measured with the stability reporter. In Fig. 5g, a one-way ANOVA was used to compare the effect of sgRNAs on nuclear KAT2A fluorescence as measured by immunofluorescence. In Fig. 5g, for each UPS gene, the mean of the two sgRNAs for each biological replicate was calculated and used as the input for statistical testing. After confirming significance, a Dunnett's multiple comparison test was performed to compare each condition with the DMSO-treated (b, c) or sgAAVS1 + DMSO (g) reference. In Fig. 5b, g, the data were normalised relative to the median nuclear KAT2A fluorescence of wild type (b), or TAF5L KO (g) cells treated with DMSO. In Fig. 5g, only eGFP-positive cells were analysed.

In Fig. 6e, a one-way ANOVA was used to compare the abundance of KAT2A-BFP for different KAT2A reporter constructs in TAF5L KO cells. After confirming significance, a Dunnett's multiple comparison test was performed to compare each condition with the reference (wild type) construct. In Fig. 6b only eGFP-positive cells were analysed. The data were normalised relative to cells treated with GSK-699 for 6 h.

In Supplementary Fig. 3b, significance was tested using a two-tailed, unpaired Welch's t-test. In Supplementary Fig. 3d, e, h, i, a one-way ANOVA was used to compare the effect of guide knockout on KAT2A-BFP (d) or mCherry (e) fluorescence, ATAC gene knockout on KAT2A abundance (h) or SAGA gene knockout on endogenous KAT2A fluorescence (i). After confirming significance, a Dunnett's multiple comparison test was performed to compare each condition with the sgAAVS1 or sgNEG reference. In i, the mean of negative control guides (sgAAVS1 and one non-targeting guide) was calculated for each replicate to generate one value per replicate. In Supplementary Fig. 3d, e, h, i, only eGFP-positive cells were analysed. In Supplementary Fig. 3b, c, the data were normalised relative to the mean KAT2A-BFP or mCherry fluorescence of transduced cells treated with DMSO. In Supplementary Fig. 3d, e, h, the data were normalised relative to the mean KAT2A-BFP or mCherry fluorescence of cells transduced with negative control guides.

In Supplementary Fig. 4b, c, significance was tested using a two-tailed, unpaired Welch's t-test. In Supplementary Fig. 4d, e, a one-way ANOVA was used to compare the effect of sgRNA knockout on KAT2A-BFP abundance. After confirming significance, a Dunnett's multiple comparison test was performed to compare each condition with the sgAAVS1 reference. In Supplementary Fig. 4b–e, only eGFP-positive cells were analysed. In b, c, the data were normalised relative to the mean KAT2A-BFP or mCherry fluorescence of transduced cells treated with DMSO. In Supplementary Fig. 4d, e, data were normalised relative to sgAAVS1 transduced cells treated with GSK-699 (100 nM) for 6 h.

In Supplementary Fig. 5, a one-way ANOVA was used to compare the effect of knockout of CORE components (b) or cDNA over-expression (e, h) on endogenous KAT2A fluorescence. After confirming significance, a Dunnett's multiple comparison test was performed to compare each condition with the wild-type reference. In Supplementary Fig. 5b, e, h, the data were normalised relative to the median nuclear KAT2A fluorescence of wild-type HAP1 cells.

In Supplementary Fig. 6a, significance was tested using a two-tailed, unpaired Welch's t-test. In Supplementary Fig. 6f, significance was used to compare KAT2B protein levels across different HAP1 cell lines. After confirming significance, a Dunnett's multiple comparison test was performed to compare each condition with the wild-type reference.

In Supplementary Fig. 7c, a one-way ANOVA was used to compare the effect of ATAC gene knockout on KAT2A-BFP abundance. After confirming significance, a Dunnett's multiple comparison test was performed to compare each condition with the sgAAVS1 reference. Only eGFP-positive cells were analysed. The data for each plot were normalised relative to sgAAVS1 transduced cells for each cell line (1 value) and cells treated with GSK-699 (100 nM) for 6 h (0 value).

In Supplementary Fig. 9, a one-way ANOVA was used to compare the effect of inhibitors (d) or sgRNA knockout (j) on KAT2A-BFP abundance as measured with the stability reporter, or the effect of guide knockout on nuclear KAT2A fluorescence (g, i) as measured by immunofluorescence. In Supplementary Fig. 9g, statistical analysis was only performed on conditions with $n = 3$ biological replicates. In Supplementary Fig. 9g, i, j, for each UPS gene, the mean of the two sgRNAs for each biological replicate was calculated and used as the input for statistical testing. After confirming significance, a Dunnett's multiple comparison test was performed to compare each condition with the DMSO-treated (d) or sgAAVS1 + DMSO (g, i, j) reference. In Supplementary Fig. 9a, the data were normalised relative to the median nuclear KAT2A fluorescence of wild-type HAP1 cells treated with DMSO. In Supplementary Fig. 9b, c, the data were normalised relative to the mean KAT2A transcript abundance of wild-type HAP1 cells. In Supplementary Fig. 9g, i, the data were normalised relative to the median nuclear KAT2A fluorescence of wild type (g) or TADA1 KO (i) cells transduced with sgAAVS1 and treated with DMSO. In Supplementary Fig. 9j, the data for each cell line were normalised relative to

the mean KAT2A-BFP abundance of GSK-699-treated cells (0 value) and sgAAVS1 transduced cells treated with DMSO (1 value). In Supplementary Fig. 9g, i, only eGFP-positive cells were analysed. In Supplementary Fig. 9j, only iRFP-positive cells were analysed.

## Reporting summary

Further information on research design is available in the Nature Portfolio Reporting Summary linked to this article.

## Data availability

CUT&RUN datasets generated in this study can be accessed on the Gene Expression Omnibus (GEO) database under the series accession number GSE300600. Mass-spectrometric raw data files as well as the data analysis output files have been deposited to the ProteomeXchange Consortium via the PRIDE partner repository with the dataset identifier PXD065443. Raw microscopy data are available and will be provided by the corresponding authors on request due to the large file sizes and large number of files involved. The number of cells analysed by microscopy in this study is listed in Supplementary Data 6. The gating strategy for the pooled CRISPR screen presented in Fig. 5e is presented in Supplementary Fig. 9e. The gating strategy for other flow cytometry analyses is presented in Supplementary Fig. 10a. Uncropped western blots are provided in Supplementary Fig. 11a–d. Source data are available for this manuscript. Source data are provided with this paper.

## Code availability

The IPython notebooks used to analyse data generated within this manuscript are available on GitHub: [https://github.com/seruggialab/Batty_et_al_Nature_Comms_2026]. The analysis notebooks have been deposited to Zenodo with the accession number [https://doi.org/10.5281/zenodo.19236627][75].

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

## Acknowledgements

The authors thank the CCRI FACS Core Unit for assistance with flow cytometry and cell sorting, the Proteomics team of the Molecular Discovery Platform at CeMM for mass-spectrometric data acquisition, the CeMM Biomedical Sequencing Facility and the Core Imaging Facility at the Medical University of Vienna for technical support. Finally, the authors thank members of the Seruggia lab for providing important feedback on this project, and Patrik Hains, Stefan Kubicek, David Haselbach and Zuzana Hodáková for insightful discussions. This research was funded in whole or in part by the Austrian Science Fund (FWF) [10.55776/P36069]. For open access purposes, the author has applied a CC BY public copyright license to any author accepted

manuscript version arising from this submission. Research in the Seruggia laboratory is supported by the Austrian Science Fund (FWF, project 10.55776/P36302), the European Research Council (ERC) under the Horizon 2020 research and innovation programme (grant agreement 947803) and under Marie Skłodowska–Curie (grant agreement 101061151). G.S.-F. is supported by the Austrian Academy of Sciences. AITHYRA and the Winter lab are supported by the Boehringer Ingelheim Foundation and the Austrian Academy of Sciences. The Winter lab is further supported by funding from the European Research Council (ERC) under the European Union's Horizon 2020 research and innovation programme (grant agreement 851478), by team KOODAC via the Cancer Grand Challenges partnership funded by Cancer Research UK (CGCATF-2023/100013), Institut National Du Cancer (INCa) and KiKa (Children Cancer Free Foundation), as well as by funding from the Austrian Science Fund (FWF, projects P5918723, P36746 and P7909) and the Vienna Science and Technology Fund (WWTF, project LS21-015).

## Author contributions

Conceptualisation: P.B. and D.S.; investigation: P.B. and D.S.; formal analysis: P.B., H.B., C.S., A.P.K. and M.A.; methodology: P.B., H.B., C.S., G.O., M.Z. and S.M.; resources: G.E.W. and D.S.; validation: P.B., H.B. and S.M.; writing—original draft: P.B. and D.S.; writing—review and editing: P.B., H.B., C.S., G.E.W., G.S.-F. and D.S.; funding acquisition: D.S.; supervision: G.E.W., G.S.-F. and D.S.

## Competing interests

G.S.-F. and G.E.W. are scientific founders and shareholders of Proxygen and Solgate Therapeutics and shareholders of Cellgate Therapeutics. G.E.W. is on the Scientific Advisory Board of Proxygen and Nexo Therapeutics. The Winter and Superti-Furga laboratories have received research funding from Pfizer. G.E.W. is an inventor on several patents and patent applications covering small molecule degraders and degrader discovery approaches. The remaining authors declare no competing interests.
