## [Transparent Peer Review file · Nature Communications]

Disruption of the SAGA CORE triggers collateral degradation of KAT2A

Corresponding Author: Dr Davide Seruggia

Version 0:

Reviewer comments:

Reviewer #1

(Remarks to the Author)

In this study, Paul Batty and colleagues from the Winter and Seruggia laboratories investigated the role of SAGA complex components in regulating the abundance and stability of KAT2a. KAT2a is the catalytic histone acetyltransferase within the SAGA complex, responsible for histone acetylation. The main aims of the study were (1) to identify critical effector proteins within the SAGA complex (as targets) and (2) to obtain mechanistic insights into the quality control of SAGA complex assembly and integrity.

To address this, the authors first generated KAT2a-level reporter cell lines (HAP cells) using a bicistronic fluorescence construct. They then performed an arrayed CRISPR screen covering all SAGA complex components. Of the six identified hits, the large pseudokinase TRRAP appears to be a false positive, while the remaining hits belong to the HAT module and the adjacent CORE module of SAGA. These results suggest that spatial proximity of complex members to KAT2a is a key determinant of its stability. The authors next validated the hit sgRNAs on endogenous KAT2a protein levels as well as on a critical chromatin modification (H3K9 acetylation). Notably, the effects were in several cases even more pronounced than in the reporter cell line. Depletion of the SAGA CORE and HAT subunits not only reduced KAT2a abundance but also impaired KAT2a chromatin association and disrupted integration of the entire SAGA complex. To dissect the kinetics of KAT2a destabilization upon perturbation of SAGA, the authors generated a degron line for one of the hits, TADA1.

Treatment with dTAG induced rapid depletion of TADA1 and led to a relatively fast destabilization of KAT2a within a few hours, consistent with a direct post-transcriptional effect (further confirmed by qPCR analysis of KAT2a mRNA levels).

Finally, the authors conducted a genome-wide CRISPR screen to identify the E3 ubiquitin ligase responsible for KAT2a degradation. In addition to general components of Cullin-RING ligases, including the neddylation machinery, depletion of UBR5 (and its associated deubiquitinase OTUD5) had the strongest stabilizing effect on KAT2a.

Overall, through elegant genetic and biochemical experiments, the authors systematically dissect the contribution of SAGA complex components to KAT2a stability and function. Particularly, the identification of the responsible E3 ligase is of high significance. Given the clarity of the results and the importance of the findings for understanding this essential chromatin-modifying complex, I strongly support publication in Nature Communications.

I have only one request for revision: since the CRISPR screen of SAGA components is a central element of the study, I believe it is essential to assess the efficiency of the sgRNAs used, by quantitative IF or Immunoblotting. This would demonstrate that the observed effects on KAT2a levels are not (or not only) due to variable knockout efficiency and would help exclude false-negative hits.

– Elmar Wolf

Reviewer #2

(Remarks to the Author)

In this study, Batty et al make use of the HAP1-Cas9 haploid cell line to investigate the contribution of individual subunits of the histone acetyltransferase complex SAGA to stability and activity of the complex, specifically of its HAT subunit and KAT2A enzyme. Through means a KAT2A stability reporter, they conduct flow cytometry-based CRISPR screens to assess requirements of individual subunits to KAT2A abundance. They verify the results on the endogenous protein in the same cell line background, also reading effects on global H3K9ac, the most well-established enzymatic readout of KAT2A / SAGA HAT activity. They elegantly show that elements of the SAGA core TADA1, TAF5L and TAF6L are required for the stability of the complex and the engagement of the HAT domain. In their absence, free KAT2A increases and progressively

undergoes proteosomal degradation through the activity of the orphan protein E3 ligase activity of UBR5.

While the results presented constitute a valuable mechanistic overview of subunit requirements for SAGA complex assembly and HAT activity, it would have been advisable for the Authors to validate the core results in independent non-haploid cell lines. Such controls would greatly strengthen the findings here reported, overcoming potential confounding effects of the haploid genome, and adding functional significance.

MAIN POINTS

1. The Authors should demonstrate that relative abundances of the different SAGA components are preserved in the haploid HAP1 cell line. This is to ensure that limiting abundances of individual components are not influencing the results by modifying the stability of the complex relative to diploid cells.
2. Similarly, the key findings should be reproduced in diploid cell lines, preferably those with differential uses of KAT2A and KAT2B as the HAT enzyme, to strengthen the relevance of the findings as to the stability and activity of the SAGA complex, independently (in principle) of the HAT activity.
3. Although the Authors show removal of KAT2A from promoters using CUT&RUN, to a degree that is broadly comparable between KAT2A KO and TAF5L loss, the differences in K3K9ac by IF are less comparable between the 2 KO. It would be important to pair chromatin analysis of KAT2A and of H3K9ac to demonstrate that the apparent loss of KAT2A is also reflected in the corresponding loss of activity and if not, why not. Is there any evidence of partial HAT subunit-containing complexes, and could these be enzymatically active with comparable target specificity even if the association at chromatin is less stable?

ADDITIONAL POINTS

1. The Authors should show that KAT2A is indeed the single or dominant SAGA HAT enzyme in this cell line. Is KAT2B/PCAF present, and if so, is it also degraded? Or is it differentially sensitive to the stability of the core?
2. While the Authors mention the participation of KAT2A in the ATAC complex and indeed highlight individual ATAC elements in Figure S3, there is no further consideration of how these may, or not, contribute, to the results. Is there ATAC complex formation and activity in the HAP1 cell lines and do the Authors observe relative enrichment in the KAT2A association with ATAC when SAGA is degraded? Is the association with the 2 complexes differentially stable? And what is the identity and significance of the ATAC subunit that is differentially under-represented in the proteomics analysis?
3. The Authors should confirm the specificity of the E3 ligase activity for KAT2A in additional, diploid, cell lines, as the conclusions have implications for the proposed future clinical targeting studies.
4. The Authors should also consider if overall proteosomal activity is comparable between HAP1 cells and diploid cell lines, as this would influence the amplitude of the effects observed.

Reviewer #3

(Remarks to the Author)

The manuscript by Paul Batty and colleagues describes how the SAGA subunit KAT2A depends on protein-protein interactions within the native complex for its incorporation, stability, and chromatin occupancy and acetylation. Using a well-designed reporter construct and orthogonal quantitative imaging, the authors identified which individual SAGA subunits and components of the ubiquitin-proteasome machinery control KAT2A stability. Specifically, their results show that KAT2A levels decline not only upon loss of HAT module subunits, but also upon disruption of non-enzymatic core components, which are essential for the integrity of the entire complex. The authors thus propose that SAGA structural integrity, specifically the TAF5L, TAF6L, and TADA1 core subunits, is essential for KAT2A, indicating that SAGA-mediated gene regulation depends on both enzymatic modules and a general architecture that prevents degradation of its components.

Overall, the experiments are well designed, carefully controlled, and clearly reported. The manuscript is well written and organized. However, in its current form, it has several flaws that must be addressed before it can be considered for publication in any journal. In particular, biological effects cannot be reliably concluded from technical replicates alone. All experiments in Figs 1-5 (except the TMT proteomics and the FACS-based CRISPR screen) have to be conducted independently at least three times, especially when drawing conclusions about the presence or absence of an effect.

In addition, once independent measurements are obtained, a one-way ANOVA should be used to compare more than two groups. A two-tailed Mann-Whitney U-test is suitable only for comparing two means, and only if these two means correspond to independent experiments. Measuring individual cells cannot be considered independent replicates because cells within the same culture dish are not statistically independent from each other.

In summary, this study primarily provides a detailed analysis of how SAGA integrity affects one of its subunits, KAT2A, building on the well-established role of SAGA core components in maintaining complex stability and function, as first described in *S. cerevisiae* over 20 years ago (e.g., Wu PYJ et al., 2002; DOI: 10.1128/MCB.22.15.5367-5379.2002). While

the experiments are carefully conducted and highly detailed, the findings appear incremental relative to existing literature on SAGA structure and function, making the work more appropriate for a specialized rather than a broad readership.

Specific comments:

- It is difficult to compare the effects of different subunits because of variable efficiency of sgRNA (eg. targeting TRRAP should induce a much faster drop of GFP-positive cells, comparable to MYC, suggesting poor efficiency (Figure S1A,B)). We recommend performing Western blot controls of sgRNA knockout efficiency, at least for core components, to validate that selected subunit are required for KAT2A stability, like TADA1, while others are not, such as SUPT20H.
- Similar to TRRAP, there are other inconsistent observations between the stability reporter and direct fluorescence measurement of KAT2A stability. For example, a TAF12 knockout has no effect on the stability reporter (Fig 1C), while nuclear fluorescence drops as much as upon TADA1 knockout (Fig S1J).
- The CUT&RUN profile of KAT2A, shown in Fig 3A, is atypical for a co-activator complex such as SAGA, which should display an enrichment upstream of the TSS and a rather poor signal downstream. Does that mirror H3K9 acetylation in these cells?
- The presence of KAT2A and TADA3 across the full range of fractions suggest that the sucrose gradient or ultracentrifugation conditions were not able to discriminate between fully assembled complexes and the monomeric forms of each subunit. While it is possible that intermediate complexes exist, they are very transient, thus difficult to detect. In addition, the existence of subcomplexes such the ADA complex in yeast has not been demonstrated in mammalian cells so far and the observation that loss of core components phenocopies loss of KAT2A, as shown here, is a strong argument against its existence. Finally, if KAT2A accumulates in lower fractions with other HAT components upon loss of core subunits, TADA3 should behave similarly, which is not apparent in Fig 3D,E.
- It is unclear how "minimal changes in the levels of ATAC and TFIID subunits" observed by TMT proteomics shows that "the reduction in protein abundancies of SAGA components is driven specifically by the acute depletion of TADA1". To test this specifically, the authors should perform immunoprecipitations of SAGA and TFIID using specific antibodies and verify, for example, that TAF12, which is shared between SAGA and TFIID, is only lost in a SAGA purification, and not from TFIID.
- In Fig 4D, KAT2A protein levels gradually reduced over time but do not reach the level of TADA1 KO cells within 24 h of dTAG treatment, as stated in the text - please rephrase.
- Page 13: "KAT2A protein levels upon proteasome inhibition was substantially higher in TAF5L KO cells compared to wild type (Fig. 5F, G, S4F, G), [...] (normalised mean KAT2A fluorescence: TAF5L KO sgAAVS1 + MG-132, 1.35 ± 0.42 ; Wild type sgAAVS1 + MG-132, 1.16 ± 0.33 , mean + S.D.)". The substantial overlap in standard deviations suggests that this conclusion may be overstated.
- Page 13: "In contrast, the magnitude of rescue was more than two-fold lower in wild type cells (Fig. 5F, G, S4F-I)". It is unclear what the authors refer to when describing a two-fold weaker effect in wild-type cells and how this supports the conclusion that UBR5 and OTUD5 are specific regulators of KAT2A degradation.
- Page 13: "Such findings are consistent with an increased abundance of low molecular weight KAT2A species in CORE KO mutants (Fig. 2D, E)". The authors probably meant Figure 3.

Reviewer #4

(Remarks to the Author)

Version 1:

Reviewer comments:

Reviewer #1

(Remarks to the Author)

The authors have substantially improved the manuscript through revision and the inclusion of additional experiments. They have fully addressed my concerns and, to the best of my knowledge, also those raised by the other reviewers. I therefore support the publication of this manuscript in Nature Communications.

Reviewer #2

(Remarks to the Author)

I thank the Authors for the extensive work that went into this revision. I am happy that the points I previously raised have been sufficiently addressed. This is an important work which I believe is now ready for publication.

Reviewer #3

(Remarks to the Author)

I have now gone through the revised manuscript and appreciate the efforts to address my concerns, particularly those related to replication and statistical analyses. The additional data and clarifications provided strengthen the work. I nevertheless have a few specific comments that I recommend the authors address:

1. Citations in the Discussion require careful verification.

For example, in the sentence:

“A high-resolution structure of the human SAGA HAT however remains unavailable⁶¹, while a recently published high-resolution HAT structure from *C. thermophilum*^{33,38}, including KAT2A, lacks the first 410 amino acids of the human KAT2A sequence, meaning the precise location of the identified degron and its potential interaction partners are not currently known.”

the cited references 33 and 38 do not appear to support the statement made. Only reference 61 seems appropriate, but it is unusual to cite a yeast structure paper to justify the absence of a high-resolution structure for the human SAGA HAT module. Moreover, although Mattoo et al. report the first structure of SAGA containing the HAT module in yeast, several of their findings are directly relevant for interpreting the phenotypes observed in human cells (e.g., the role of TADA1). It would therefore be valuable to integrate and acknowledge this work more explicitly, as well as previous work about SAGA structural integrity in yeast and flies more generally, where appropriate.

2. The manuscript would benefit from careful editing by a fluent English speaker.

Some sentences are difficult to understand or syntactically incorrect. For instance:

“suggesting that in clonally derived KAT2A KO cells that compensatory mechanisms exist that can to a large extent retain H3K9ac at promoters.”

Several passages would benefit from restructuring for clarity and readability.

3. Figure S6d does not support the claim of “transcriptional induction” of KAT2B.

The current data in Figure S6d do not show transcriptional upregulation of KAT2B upon loss of KAT2A, contrary to what is stated on page 9. This point should be revised or qualified accordingly.

Reviewer #4

(Remarks to the Author)

Rebuttal letter concerning “*Disruption of the SAGA CORE triggers collateral degradation of KAT2A*”, *Nature Communications*

Reviewer #1 (Remarks to the Author):

In this study, Paul Batty and colleagues from the Winter and Seruggia laboratories investigated the role of SAGA complex components in regulating the abundance and stability of KAT2a. KAT2a is the catalytic histone acetyltransferase within the SAGA complex, responsible for histone acetylation. The main aims of the study were (1) to identify critical effector proteins within the SAGA complex (as targets) and (2) to obtain mechanistic insights into the quality control of SAGA complex assembly and integrity. To address this, the authors first generated KAT2a-level reporter cell lines (HAP cells) using a bicistronic fluorescence construct. They then performed an arrayed CRISPR screen covering all SAGA complex components. Of the six identified hits, the large pseudokinase TRRAP appears to be a false positive, while the remaining hits belong to the HAT module and the adjacent CORE module of SAGA. These results suggest that spatial proximity of complex members to KAT2a is a key determinant of its stability. The authors next validated the hit sgRNAs on endogenous KAT2a protein levels as well as on a critical chromatin modification (H3K9 acetylation). Notably, the effects were in several cases even more pronounced than in the reporter cell line. Depletion of the SAGA CORE and HAT subunits not only reduced KAT2a abundance but also impaired KAT2a chromatin association and disrupted integration of the entire SAGA complex. To dissect the kinetics of KAT2a destabilization upon perturbation of SAGA, the authors generated a degron line for one of the hits, TADA1. Treatment with dTAG induced rapid depletion of TADA1 and led to a relatively fast destabilization of KAT2a within a few hours, consistent with a direct post-transcriptional effect (further confirmed by qPCR analysis of KAT2a mRNA levels). Finally, the authors conducted a genome-wide CRISPR screen to identify the E3 ubiquitin ligase responsible for KAT2a degradation. In addition to general components of Cullin-RING ligases, including the neddylation machinery, depletion of UBR5 (and its associated deubiquitinase OTUD5) had the strongest stabilizing effect on KAT2a. Overall, through elegant genetic and biochemical experiments, the authors systematically dissect the contribution of SAGA complex components to KAT2a stability and function. Particularly, the identification of the responsible E3 ligase is of high significance. Given the clarity of the results and the importance of the findings for understanding this essential chromatin-modifying complex, I strongly support publication in *Nature Communications*.

I have only one request for revision: since the CRISPR screen of SAGA components is a central element of the study, I believe it is essential to assess the efficiency of the sgRNAs used, by quantitative IF or Immunoblotting. This would demonstrate that the observed effects on KAT2a levels are not (or not only) due to variable knockout efficiency and would help exclude false-negative hits.

– Elmar Wolf

We thank the reviewer for underlining the clarity of results and relevance of the findings of our work. Given its relevance, in the revised version we expanded on the mechanism of UBR5-mediated KAT2A degradation and identified a degron motif responsible for KAT2A degradation (new Fig. 6).

To address the specific reviewer's request, in the absence of antibodies for all 20 components of SAGA, we have confirmed gRNA editing using Sanger sequencing. All gRNAs used result in high and comparable editing (**Suppl. Fig. 1**).

Reviewer #2 (Remarks to the Author)

In this study, Batty et al make use of the HAP1-Cas9 haploid cell line to investigate the contribution of individual subunits of the histone acetyltransferase complex SAGA to stability and activity of the complex, specifically of its HAT subunit and KAT2A enzyme. Through means a KAT2A stability reporter, they conduct flow cytometry-based CRISPR screens to assess requirements of individual subunits to KAT2A abundance. They verify the results on the endogenous protein in the same cell line background, also reading effects on global H3K9ac, the most well-established enzymatic readout of KAT2A / SAGA HAT activity. They elegantly show that elements of the SAGA core TADA1, TAF5L and TAF6L are required for the stability of the complex and the engagement of the HAT domain. In their absence, free KAT2A increases and progressively undergoes proteosomal degradation through the activity of the orphan protein E3 ligase activity of UBR5. While the results presented constitute a valuable mechanistic overview of subunit requirements for SAGA complex assembly and HAT activity, it would have been advisable for the Authors to validate the core results in independent non-haploid cell lines. Such controls would greatly strengthen the findings here reported, overcoming potential confounding effects of the haploid genome, and adding functional significance.

We thank the reviewer for appreciating the value of our mechanistic work. We address individual points below, including validation of core results in independent non-haploid cell lines. Comments from the reviewer motivated us to perform additional experiments, leading to interesting new insights regarding the association of KAT2A with ATAC and the differential sensitivity of paralogues KAT2A and KAT2B towards degradation upon perturbation of the SAGA CORE.

MAIN POINTS

1. The Authors should demonstrate that relative abundances of the different SAGA components are preserved in the haploid HAP1 cell line. This is to ensure that limiting abundances of individual components are not influencing the results by modifying the stability of the complex relative to diploid cells.

HAP1 cells were successfully used to characterize other large multi-protein chromatin regulator complexes such as the SWI/SNF (Schick et al, Nat Genet 2019, doi: 10.1038/s41588-019-0477-9; Schick et al. Nat Genet 2021, doi: 10.1038/s41588-021-00777-3), which motivated our choice of using such a model for investigating aspects of the SAGA complex. To address the reviewer's point, we retrieved proteomics data obtained from a set of diploid cells (PRIDE identifier: PXD051747) and compared relative abundance of detectable SAGA components in wild-type HAP1 cells. We observed that the relative abundance of SAGA components is comparable across cell lines, including HAP1 (**Reviewer Fig. 1**). We noticed that HAP1 cells express higher levels of USP22, which is an interesting observation, that does not however affect any of our findings. Please note that we were unable to identify unique peptides for TADA2B, ATXN7, SUPT7L and SUPT20H in our dataset, as already shown in **Fig. 4e**. Therefore, we exclude that limiting abundance of individual components could influence the results in haploid cells, as demonstrated in experiments using leukemia cell lines (see answer to the next comment as well).

2. Similarly, the key findings should be reproduced in diploid cell lines, preferably those with differential uses of KAT2A and KAT2B as the HAT enzyme, to strengthen the relevance of the findings as to the stability and activity of the SAGA complex, independently (in principle) of the HAT activity.

We thank the reviewer for this question. To address if SAGA CORE disruption reduces KAT2A abundance in cell types other than HAP1, we used two leukemia cell lines, MOLM-13 and NALM6, expressing low and high levels of KAT2B, respectively (Suppl. Fig. 4a), along with Cas9. First, we introduced the KAT2A fluorescent reporter, that we validated using the KAT2A/KAT2B PROTAC (Suppl. Fig. 4b, c). Next, we proceeded with gRNA transductions, targeting either SAGA CORE components or AAVS1. We observed destabilisation of KAT2A upon targeting the SAGA CORE components TADA1 and TAF5L, confirming results from HAP1 cells (Suppl. Fig. 4d, e).

Supplementary Figure 4 (new): a, Expression of KAT2B in MOLM13 and NALM6 cells (from Depmap). **b-c**, Validation of the KAT2A stability reporter in MOLM-13 (**b**) and NALM6 cells (**c**). Abundance of KAT2A-BFP upon KO of TADA1, TAF5L and TAF6L in MOLM-13 (**d**) and NALM6 cells (**e**).

3. Although the Authors show removal of KAT2A from promoters using CUT&RUN, to a degree that is broadly comparable between KAT2A KO and TAF5L loss, the differences in H3K9ac by IF are less comparable between the 2 KO. It would be important to pair chromatin analysis of KAT2A and of H3K9ac to demonstrate that the apparent loss of KAT2A is also reflected in the corresponding loss of activity and if not, why not. Is there any evidence of partial HAT subunit-containing complexes, and could these be enzymatically active with comparable target specificity even if the association at chromatin is less stable?

We thank the reviewer for this observation. To adequately compare KAT2A and H3K9ac on chromatin, we performed CUT&RUN for H3K9ac in KO clones. We observed a reduction in H3K9ac signal at TSSs in TAF5L KO cells, consistent with loss of KAT2A on chromatin (**Fig. 2g, h**). A similar drop in H3K9ac signal was also observed in TAF6L KO cells. We (and others) observed that prolonged depletion of KAT2A induces a compensatory mechanism that upregulates KAT2B (**Fig. 2i, j, and Suppl. Fig. 6d-f**; see answer to *Additional point 1*, below), rendering H3K9ac levels in KAT2A KO indistinguishable from wild type cells. These experiments highlight that targeting specific CORE components reduces the levels of H3K9ac, while prolonged inactivation of KAT2A does not. Importantly, KO of the SAGA CORE component TAF5L does not induce the upregulation of KAT2B (**Fig. 2i, j**).

Regarding the existence of partial HAT-subunit containing complexes, while SAGA HAT subcomplexes have been reported in yeast (Grant et al. *Genes Dev* 1997, 10.1101/gad.11.13.1640), there is no evidence for their existence in mammalian cells. In fact, our data suggests that HAT subunits are exposed to proteasomal degradation and hence inactive when separated from SAGA (**Fig. 4e, Fig. 5**).

Figure 2 (new) e, KAT2A CUT&Run signal pile-up at TSSs in wild type, KAT2A KO and TAF5L KO HAP1 cells. f, KAT2A binding at the eEF1A1 locus. g, H3K9ac CUT&Run signal pile-up at TSSs in wild type, KAT2A, TAF5L and TAF6L KO HAP1 cells. h, Overlay of H3K9ac profiles at TSSs. i, j, Immunofluorescence of KAT2B in wild type, KAT2A KO and TAF5L KO HAP1 cells.

ADDITIONAL POINTS

1. The Authors should show that KAT2A is indeed the single or dominant SAGA HAT enzyme in this cell line. Is KAT2B/PCAF present, and if so, is it also degraded? Or is it differentially sensitive to the stability of the core?

We thank the reviewer for raising this point, that led to new interesting results. Our new data show that KAT2A is the dominant SAGA HAT in HAP1 cells, with KAT2B being weakly detected at the RNA and protein level (**Suppl. Fig. 6a, f**). We report that while KAT2B expression is very low in HAP1 cells, KO of KAT2A induces a strong upregulation of KAT2B (**Suppl. Fig. 6d-f**). This finding is generalisable to other cell lines, as shown by the Stegmaier lab in the context of neuroblastoma (doi:10.1126/sciadv.adm9449). Since KAT2B levels in wild type cells are too low to test its sensitivity to CORE perturbations, we addressed this question in KAT2A KO cells, that in turn express high levels of KAT2B. Curiously, differently from KAT2A, KAT2B levels are unaffected by KO of SAGA CORE components TAF5L, TAF6L and TADA, suggesting a paralogue-specific degradation pathway (**Fig. 6a, b**).

To better understand the differences between proteostatic regulation of the two paralogues, we looked at protein sequence conservation. While KAT2A and KAT2B share ~70% of sequence homology, the two proteins differ at their N-terminus (**Fig. 6c, d**) and intriguingly, UBR5 (Ubiquitin

Protein Ligase E3 Component N-Recognin 5), the E3 ligase we identified as responsible for degrading solitary KAT2A (Fig. 5), tends to recognize residues at disordered domains, typically rich in either basic or hydrophobic residues, such as the N-terminus of KAT2A. Thus, we identified and tested the effect of mutating a putative degron motif at the KAT2A N-terminus, absent in KAT2B, as well as other predicted degrons. We introduced mutations in the KAT2A fluorescent reporter and tested if the degron at the KAT2A N-terminus is involved in KAT2A degradation. Indeed, mutagenesis of the N-term degron rescues KAT2A stability in TAF5L KO cells, while inactivation of other three predicted degrons does not influence KAT2A levels (Fig. 6d, e).

We compiled data about KAT2B upregulation in Suppl. Fig. 6 and included experiments to test candidate KAT2A degron sequences in Fig. 6.

Supplementary Figure 6, (new). **a**, RT-qPCR to measure relative expression of KAT2A and KAT2B in wild type HAP1 cells. **b,c**, Immunofluorescence of KAT2A in wild type parental HAP1^{Cas9} cells and in two KAT2A KO clones. **d**, Relative mRNA levels of KAT2B in wild type and KAT2A KO cells. **e,f** Representative images and quantification of KAT2B levels in wild type parental HAP1^{Cas9} cells, in cells treated with the KAT2A/KAT2B PROTAC and in two distinct KAT2A KO clones.

Figure 6 (new). **a,b**, Immunofluorescence of KAT2B in KAT2A KO cells upon targeting TADA1, TAF5L or TAF6L. Levels of KAT2B are unaffected when the SAGA core is perturbed. **c**, Map of KAT2A and its predicted degens. Sequence homology with KAT2B is shown as a purple box. **d**, Amino acid sequences of the four predicted KAT2A degens. Mutations introduced into the KAT2A reporter are shown in red. **e**, Normalized KAT2A-BFP levels of different KAT2A mutants in TAF5L KO HAP1 cells. Mutagenesis of the UBR5 degen uniquely found within the KAT2A N-terminus elevates KAT2A levels in TAF5L KO cells.

While the Authors mention the participation of KAT2A in the ATAC complex and indeed highlight individual ATAC elements in Figure S3, there is no further consideration of how these may, or not, contribute, to the results. Is there ATAC complex formation and activity in the HAP1 cell lines and do the Authors observe relative enrichment in the KAT2A association with ATAC when SAGA is degraded? Is the association with the 2 complexes differentially stable?

This is an interesting point, that we addressed by targeting components of the ATAC complex in wild type cells or in cells lacking a CORE component (TAF5L KO), where the integrity of SAGA is perturbed. In wild type cells, KAT2A levels (BFP/mCherry ratio) are not altered by perturbations of proteins that are unique members of ATAC (Suppl. Fig. 3g, h). However, when we target ATAC components in TAF5L KO cells, where the integrity of SAGA is perturbed, we observe an additive effect on the loss of KAT2A stability (Suppl. Fig. 7c). This suggests that, while it mostly

and primarily associates with SAGA, when this becomes unavailable, KAT2A is able to associate with ATAC.

Supplementary Figure 3g, Components of SAGA and ATAC complexes; shared components are marked by an asterix. **h**, Normalized KAT2A-BFP levels upon KO of SAGA CORE, HAT and ATAC components in wild type HAP1^{Cas9} cells.

Supplementary Figure 7c: Normalized KAT2A-BFP levels upon KO of SAGA CORE, HAT and ATAC components in TAF5L KO HAP1^{Cas9} cells.

And what is the identity and significance of the ATAC subunit that is differentially under-represented in the proteomics analysis?

The ATAC component downregulated after 1, 6, and 24 h of TADA1 depletion is MBIP. The plots in **Suppl. Fig. 8b-d** have been annotated accordingly. Although MBIP levels were reduced upon TADA1 depletion in our proteomics dataset, knockout of MBIP does not affect KAT2A stability (**Suppl. Fig. 3g**; **Suppl. Fig. 7c**), suggesting that the two proteins might not physically interact.

3. The Authors should confirm the specificity of the E3 ligase activity for KAT2A in additional, diploid, cell lines, as the conclusions have implications for the proposed future clinical targeting studies.

We agree with the reviewer on the necessity of testing that UBR5-mediated degradation of KAT2A translates to cancer cell lines. To address this point, we knocked out UBR5 or OTUD5 in the AML cell lines MOLM-13 and MV4-11 and measured KAT2A stability using the KAT2A fluorescence reporter. We observed an increase in KAT2A abundance upon KO of UBR5 or OTUD5, confirming findings in HAP1 cells. We include this experiment as a new panel in **Suppl. Fig. 9j**.

Supplementary Fig. 9j (new). Normalized KAT2A-BFP levels in MOLM-13^{Cas9} and MV4-11^{Cas9} cells transduced with gRNAs targeting AAVS1, OTUD5 or UBR5.

4. The Authors should also consider if overall proteasomal activity is comparable between HAP1 cells and diploid cell lines, as this would influence the amplitude of the effects observed.

To quantitatively probe for overall proteasomal activity, we performed a time course using GSK-699, a proteolysis targeting chimera (PROTAC) composed of a KAT2A bromodomain ligand fused to pomalidomide, and used our KAT2A fluorescence reporter to measure KAT2A abundance as a proxy for proteasomal activity. The kinetics and amplitude of KAT2A degradation following GSK-699 treatment are overall comparable across the tested cell lines, indicating similar overall activity of CRBN-mediated proteasomal degradation. Data is displayed in **Reviewer Figure 2**.

Reviewer Figure 2: Kinetics of KAT2A degradation using 100 nM GSK-699 in different cell lines. GSK699 is a PROTAC mediating proteasomal degradation of KAT2A via CRBN engagement. Therefore, the kinetics of KAT2A abundance upon GSK-699 treatment indicate the overall proteasomal activity.

Reviewer #3 (Remarks to the Author)

The manuscript by Paul Batty and colleagues describes how the SAGA subunit KAT2A depends on protein-protein interactions within the native complex for its incorporation, stability, and chromatin occupancy and acetylation. Using a well-designed reporter construct and orthogonal quantitative imaging, the authors identified which individual SAGA subunits and components of the ubiquitin-proteasome machinery control KAT2A stability. Specifically, their results show that KAT2A levels decline not only upon loss of HAT module subunits, but also upon disruption of non-enzymatic core components, which are essential for the integrity of the entire complex. The authors thus propose that SAGA structural integrity, specifically the TAF5L, TAF6L, and TADA1 core subunits, is essential for KAT2A, indicating that SAGA-mediated gene regulation depends on both enzymatic modules and a general architecture that prevents degradation of its components. Overall, the experiments are well designed, carefully controlled, and clearly reported. The manuscript is well written and organized. However, in its current form, it has several flaws that must be addressed before it can be considered for publication in any journal. In particular, biological effects cannot be reliably concluded from technical replicates alone. All experiments in Figs 1-5 (except the TMT proteomics and the FACS-based CRISPR screen) have to be conducted independently at least three times, especially when drawing conclusions about the presence or absence of an effect. In addition, once independent measurements are obtained, a one-way ANOVA should be used to compare more than two groups. A two-tailed Mann-Whitney U-test is suitable only for comparing two means, and only if these two means correspond to independent experiments. Measuring individual cells cannot be considered independent replicates because cells within the same culture dish are not statistically independent from each other. In summary, this study primarily provides a detailed analysis of how SAGA integrity affects one of its subunits, KAT2A, building on the well-established role of SAGA core components in maintaining complex stability and function, as first described in *S. cerevisiae* over 20 years ago (e.g., Wu PYJ et al., 2002; DOI: 10.1128/MCB.22.15.5367–5379.2002). While the experiments are carefully conducted and highly detailed, the findings appear incremental relative to existing literature on SAGA structure and function, making the work more appropriate for a specialized rather than a broad readership.

We thank the reviewer for recognizing the quality of our work, the experimental design, the choice of orthogonal validations and the clarity of the manuscript. While the modular nature of the SAGA complex was already appreciated in yeast, here we report for the first time a pathway for solitary KAT2A degradation in human cells and identified the responsible E3 ligase. Furthermore, in the revised manuscript we tested the contribution of ATAC components in buffering solitary KAT2A when SAGA is perturbed (**Suppl. Fig 7c**). Finally, we revealed differences in the proteostasis of the paralogues KAT2A and KAT2B (**Suppl. Fig. 6**) and identified a KAT2A-specific degron motif (**Fig. 6**) within the N-terminus of KAT2A. We believe this contributes to elevating the novelty of our work.

We clarify that experiments were conducted in at least biological triplicate (as reported in the respective figure legends), with the exception of **Fig. 5f** (initially two replicates), now amended in the revised manuscript to three repeats. In the revised manuscript, we provide data plotted as

suggested by the reviewer, comparing means of biological triplicates, and apply the recommended statistical test. For immunofluorescence data, we now show the mean of cells from each biological replicate (black dots), with a red line at the mean of three (or more) biological replicates, while data points of all individual cells are represented as an overlaid scatterplot. Statistics are exclusively performed on biological replicates.

Assays, number of biological replicates and statistical tests used are summarized in the table below; detailed information is provided in the respective figure legends.

Panel	n. of biological reps	Statistical test
FACS-based stability reporter assay (Fig. 1c; S3h, S7c)	n = 6	One-way ANOVA
KAT2A quantification by IF (Fig. 1e; Fig. S3i)	n = 6	One-way ANOVA
KAT2A quantification by IF (Fig. 2b)	n = 3-7	One-way ANOVA
H3K9ac quantification by IF (Fig. 2d)	n = 3	One-way ANOVA
HA quantification by IF (Fig. 4b)	n = 3	One-way ANOVA
KAT2A quantification by IF (Fig. 4d)	n = 3	One-way ANOVA
KAT2A quantification by IF (Fig. 5b, g; Fig. S9g, i)	n = 3	One-way ANOVA
FACS-based stability reporter assay (Fig. 5c)	n = 4	One-way ANOVA
Pooled CRISPR screen (Fig. 5e)	n = 3	Values calculated using MAGeCK
FACS-based stability reporter assay (Fig. 9d, j)	n = 3	One-way ANOVA
KAT2B quantification by IF (Fig. 2j, Fig. 6b; Fig. S6f)	n = 3	One-way ANOVA
FACS-based stability reporter assay (Fig. 6e)	n = 4	One-way ANOVA

Specific comments:

- It is difficult to compare the effects of different subunits because of variable efficiency of sgRNA (eg. targeting TRRAP should induce a much faster drop of GFP-positive cells, comparable to MYC, suggesting poor efficiency (Figure S1A,B)). We recommend performing Western blot controls of sgRNA knockout efficiency, at least for core components, to validate that selected subunit are required for KAT2A stability, like TADA1, while others are not, such as SUPT20H.

Due to a lack of suitable commercial antibodies for several CORE and SAGA components, we assessed the editing efficiency of sgRNAs by Sanger sequencing. Data is included in new **Supplementary Fig. 1**. This analysis confirms high and comparable editing efficiencies of gRNAs across the screen.

- Similar to TRRAP, there are other inconsistent observations between the stability reporter and direct fluorescence measurement of KAT2A stability. For example, a TAF12 knockout has no effect on the stability reporter (Fig 1C), while nuclear fluorescence drops as much as upon TADA1 knockout (Fig S1J).

We thank the reviewer for pointing out this discrepancy. While the stability reporter allows fast and inexpensive estimation of KAT2A abundance by FACS upon genetic perturbations, quantitative microscopy of endogenous KAT2A reflects the abundance of KAT2A more accurately. We further addressed the limitations of the fluorescent reporter in evaluating the effect of TRRAP KO in **Suppl. Fig.3f**:

“However, closer inspection of the 2D FACS plots for cells transduced with guides against *TRRAP* showed that these cells had an increase in mCherry signal compared to control cells, rather than a decrease in KAT2A-BFP fluorescence (**Fig. S3f**), suggesting an artefact of the reporter leads to a reduced KAT2A-BFP/mCherry ratio when targeting *TRRAP*.”

Discrepancies between the two assays when evaluating the effect of TAF12 knockout might be due to the essentiality of this factor. Nevertheless, we followed up on TADA1, TAF5L and TAF6L, unique components of the SAGA complex, that score positive in both the FACS arrayed screen and the quantitative microscopy validation experiment. We further comment on TAF12 and its structural relationship with TADA1 in the revised manuscript:

“In addition, TAF12, an essential gene and shared component of the SAGA CORE and TFIID complexes, which did not score as a hit in our arrayed screen, also resulted in a significant reduction in KAT2A protein levels (**Fig. S3i**), equivalent to that observed upon knockout of TADA1. Indeed, consistent with this finding, TAF12 and TADA1 are known to fold co-translationally⁷ and interact via their histone fold domains, forming a ‘handshake’^{20,35}, consistent with a common mechanism of action upon their knockout leading to reduced KAT2A abundance.”

- The CUT&RUN profile of KAT2A, shown in Fig 3A, is atypical for a co-activator complex such as SAGA, which should display an enrichment upstream of the TSS and a rather poor signal downstream. Does that mirror H3K9 acetylation in these cells?

To address this point we performed CUT&RUN for H3K9ac in HAP1 cells, including WT, KAT2A KO, TAF5L KO and TAF6L KO (see **Fig. 2g, h**) and compared the profiles of KAT2A and H3K9ac in wild type cells. In agreement with the KAT2A CUT&Run, the H3K9ac profile around TSS shows a shoulder of higher signal at TSS ~ +1kb, which mirrors the profile of KAT2A CUT&Run (**Reviewer Figure 3**).

Reviewer Figure 3. a, KAT2A CUT&Run signal pile-up at TSS regions (-2kb, +2kb). **b**, H3K9ac CUT&Run signal pile-up at TSS regions (-2kb, +2kb).

- The presence of KAT2A and TADA3 across the full range of fractions suggest that the sucrose gradient or ultracentrifugation conditions were not able to discriminate between fully assembled complexes and the monomeric forms of each subunit. While it is possible that intermediate complexes exist, they are very transient, thus difficult to detect. In addition, the existence of subcomplexes such the ADA complex in yeast has not been demonstrated in mammalian cells so far and the observation that loss of core components phenocopies loss of KAT2A, as shown here, is a strong argument against its existence. Finally, if KAT2A accumulates in lower fractions with other HAT components upon loss of core subunits, TADA3 should behave similarly, which is not apparent in Fig 3D,E.

While we performed sucrose fractionations to the best of our abilities, loading all consecutive fractions on PAGE, we cannot exclude that partial complex disassembly happened during sample preparation. Furthermore, both TADA3 and KAT2A are part of complexes other than SAGA in mammalian cells (such as ATAC), with different compositions and molecular weights which might result in bands of intermediate molecular weight. Nevertheless, KAT2A and TADA3 are very clearly depleted from high molecular weight fractions in CORE mutants compared to wild-type control. To corroborate these findings, we performed a rescue experiment and observed that overexpression of murine Tada1 in TADA1 KO cells restores KAT2A and TADA3 to high molecular weight fractions, with largely the same pattern observed in WT cells (**Suppl. Fig. 7a, b**).

Regarding the accumulation of KAT2A in lower fractions, we decided to remove that statement from the manuscript, and removed arrows highlighting the low molecular weight fractions from the figure, as we cannot properly quantify such accumulation. Instead, we focused on loss of high-molecular weight fractions in KOs (**Fig. 3a, b**), that is restored by overexpression of murine TADA1 (**Suppl. Fig. 7a, b**). We believe this rescue experiment further strengthens the argument that SAGA assembly is perturbed in the absence of certain CORE components.

Supplementary Figure 7a,b (new). Sucrose gradient fractionation on wild type cells, in TADA1 KO cells, and in TADA1 KO cells stably expressing a Tada1 cDNA expression construct. Fractionated protein was blotted and stained with KAT2A (a) and TADA3 (b) antibodies.

- It is unclear how "minimal changes in the levels of ATAC and TFIID subunits" observed by TMT proteomics shows that "the reduction in protein abundancies of SAGA components is driven specifically by the acute depletion of TADA1". To test this specifically, the authors should perform immunoprecipitations of SAGA and TFIID using specific antibodies and verify, for example, that TAF12, which is shared between SAGA and TFIID, is only lost in a SAGA purification, and not from TFIID.

We agree with the reviewer that this might be an overstatement not supported by the provided proteomics dataset. We removed the statement from the revised manuscript and revised the concluding sentence of the paragraph:

"The abundance of components from other protein complexes that share subunits with SAGA was however largely unchanged, with minimal changes in the levels of ATAC and TFIID subunits (**Fig. S3b-e**). Thus, acute depletion of TADA1 results not only in reduced KAT2A protein levels, but also the progressive loss of numerous other components of SAGA upon prolonged TADA1 degradation."

- In Fig 4D, KAT2A protein levels gradually reduced over time but do not reach the level of TADA1 KO cells within 24 h of dTAG treatment, as stated in the text - please rephrase.

The relevant sentence has been modified accordingly in the revised manuscript (changes in red):

"KAT2A protein levels gradually reduced over time, reaching **close to** the level of TADA1 KO cells within 24 h of dTAG^V-1 treatment (**Fig. 4c, d**)."

- Page 13: "KAT2A protein levels upon proteasome inhibition was substantially higher in TAF5L KO cells compared to wild type (Fig. 5F, G, S4F, G), [...] (normalised mean KAT2A fluorescence: TAF5L KO sgAAVS1 + MG-132, 1.35 ± 0.42 ; Wild type sgAAVS1 + MG-132, 1.16 ± 0.33 , mean + S.D.)". The substantial overlap in standard deviations suggests that this conclusion may be overstated.

With the new analysis suggested by the reviewer and with the addition of a third biological replicate, we confirm that proteasomal inhibition rescues KAT2A levels to a higher extent in TAF5L KO cells (1.41 ± 0.13 , **Fig. 5g**), compared to wild type cells (1.16 ± 0.02 , **Supp Fig. 9g**).

Figure 5g. Quantification of nuclear KAT2A fluorescence in TAF5L KO cells following knockout of UPS genes using two independent sgRNAs. Grey dots correspond to the mean of individual nuclei; black dots or triangles correspond to the mean signal for each sgRNA, red bars indicate the mean of biological replicates.

Supplementary Figure 9g. Quantification of nuclear KAT2A fluorescence in wild type HAP1 cells following knockout of UPS genes using two independent sgRNAs. Grey dots correspond to the mean of individual nuclei; black dots or triangles correspond to the mean signal for each sgRNA; red bars indicate the mean of biological replicates.

- Page 13: "In contrast, the magnitude of rescue was more than two-fold lower in wild type cells (Fig. 5F, G, S4F-I)". It is unclear what the authors refer to when describing a two-fold weaker effect in wild-type cells and how this supports the conclusion that UBR5 and OTUD5 are specific regulators of KAT2A degradation.

We rephrased to increase clarity:

In contrast, **although knockout of OTUD5 and UBR5 also increased KAT2A levels in wild type cells, the magnitude of rescue was substantially lower compared to CORE KO cells S9f, g**, implicating UBR5 and OTUD5 as **important specific** regulators of KAT2A degradation **specifically** when the SAGA CORE is perturbed.

- Page 13: "Such findings are consistent with an increased abundance of low molecular weight KAT2A species in CORE KO mutants (Fig. 2D, E)". The authors probably meant Figure 3.

We apologise for the mistake. All figure callouts have been revised and updated throughout the manuscript.

Reviewer #4 (Remarks to the Author):

We thank Reviewer 4 for their contribution to the evaluation of our manuscript.

Point-by-point rebuttal of additional reviewer comments

Reviewer #1 (Remarks to the Author):

The authors have substantially improved the manuscript through revision and the inclusion of additional experiments. They have fully addressed my concerns and, to the best of my knowledge, also those raised by the other reviewers. I therefore support the publication of this manuscript in Nature Communications.

We thank the Reviewer for their positive assessment.

Reviewer #2 (Remarks to the Author):

I thank the Authors for the extensive work that went into this revision. I am happy that the points I previously raised have been sufficiently addressed. This is an important work which I believe is now ready for publication.

We thank the Reviewer for their positive assessment.

Reviewer #3 (Remarks to the Author):

I have now gone through the revised manuscript and appreciate the efforts to address my concerns, particularly those related to replication and statistical analyses. The additional data and clarifications provided strengthen the work. I nevertheless have a few specific comments that I recommend the authors address:

1. Citations in the Discussion require careful verification. For example, in the sentence: “A high-resolution structure of the human SAGA HAT however remains unavailable⁶¹, while a recently published high resolution HAT structure from *C. thermophilum*^{33, 38}, including KAT2A, lacks the first 410 amino acids of the human KAT2A sequence, meaning the precise location of the identified degron and its potential interaction partners are not currently known.” the cited references 33 and 38 do not appear to support the statement made. Only reference 61 seems appropriate, but it is unusual to cite a yeast structure paper to justify the absence of a high resolution structure for the human SAGA HAT module. Moreover, although Mattoo et al. report the first structure of SAGA containing the HAT module in yeast, several of their findings are directly relevant for interpreting the phenotypes observed in human cells (e.g., the role of TADA1). It would therefore be valuable to integrate and acknowledge this work more explicitly, as well as previous work about SAGA structural integrity in yeast and flies more generally, where appropriate.

We apologise for this mistake and thank the reviewer for bringing this to our attention. Citing references 33 and 38 in the context of the SAGA protein structure was inadvertent, due to an error in the reference management app. We have included the correct

references and carefully checked the referencing in the rest of the text to ensure that the call outs are correct.

2. The manuscript would benefit from careful editing by a fluent English speaker. Some sentences are difficult to understand or syntactically incorrect. For instance: “suggesting that in clonally derived KAT2A KO cells that compensatory mechanisms exist that can to a large extent retain H3K9ac at promoters.” Several passages would benefit from restructuring for clarity and readability.

We thank the reviewer for this suggestion. We have reviewed the manuscript once more for clarity and readability, including the sentence highlighted by the reviewer. The manuscript was written and edited by native English speakers, and we believe the text is grammatically correct. Nevertheless, we have considered the reviewer’s suggestion and taken steps to ensure that the wording throughout the manuscript is as clear as possible, amending sentences where necessary. The specific sentence mentioned by the reviewer has been changed as follows to improve clarity:

However, H3K9ac signal in KAT2A KO cells was largely unchanged compared to wild type (**Fig. 2g, h**), suggesting that compensatory mechanisms exist in KAT2A KO cells that can largely retain H3K9ac at promoters.

3. Figure S6d does not support the claim of “transcriptional induction” of KAT2B. The current data in Figure S6d do not show transcriptional upregulation of KAT2B upon loss of KAT2A, contrary to what is stated on page 9. This point should be revised or qualified accordingly.

We apologise for this oversight and have amended the sentence referring to upregulation of KAT2B accordingly. We now refer only to KAT2A upregulation at the protein level, and state that KAT2B transcript levels were not significantly increased.

Consistent with these findings, although KAT2B is normally only weakly expressed in HAP1 cells (**Fig. S6a**), (**Fig. S6b-f**), KAT2B protein levels were strongly upregulated in KAT2A KO cells, although KAT2B transcript levels were not significantly increased (**Fig. S6b-f**), suggesting that in the long-term absence of KAT2A, KAT2B can to a large extent compensate for its absence.”

Reviewer #4 (Remarks to the Author):

We thank the Reviewer for their positive assessment.